# INTERNET-AUGMENTED LANGUAGE MODELS THROUGH FEW-SHOT PROMPTING FOR OPEN-DOMAIN QUESTION ANSWERING

## ABSTRACT

In this work, we aim to capitalize on the unique few-shot capabilities of large-scale language models (LLMs) to overcome some of their challenges with respect to grounding to factual and up-to-date information. Motivated by semi-parametric language models (LMs), which ground their decisions in external retrieved evidence, we use few-shot prompting to learn to condition LMs on information returned from the web using Google Search, a broad and constantly updated knowledge source. Our approach does not involve fine-tuning or learning additional parameters, thus making it applicable to any LM, offering therefore a strong baseline. Indeed, we find that LMs conditioned on the web surpass performance of closed-book models of similar, or even larger, model sizes in open-domain question answering. Finally, we find that increasing the inference-time compute of models, achieved via using multiple retrieved evidences to generate multiple answers followed by a reranking stage that uses scores generated by the same LMs, leads to better performance and alleviates lower performance of smaller few-shot LMs. All in all, our findings suggest that it might be beneficial to slow down the race towards the biggest model and instead shift attention towards finding more effective ways to use models, including but not limited to, better prompting or increasing inference-time compute.

## 1 INTRODUCTION

Undoubtedly, large-scale language models (LLMs) present a breakthrough for language research, particularly for their state-of-the-art language modeling results (Radford et al., 2019; Rae et al., 2021) and impressive generative capabilities. Above all, increasing scale has made few-shot learning a defining new paradigm for language models (LMs). Due to the versatility of prompting, these models can now be quickly adapted using only a handful of examples to perform tasks ranging from question answering and numeric reasoning to creative writing (Brown et al., 2020). All these considerations place few-shot LLMs at an excellent position to be used as building blocks for open-ended and "in the wild" user interactions.

Despite these successes, few-shot LLMs still lack a key ingredient; they are susceptible to hallucinations (Maynez et al., 2020) caused by incorrect retrieval of knowledge stored in their weights or due to the model having incomplete or outdated knowledge. As for many user interactions we expect factuality to play an important role, it is imperative to find ways to keep LLMs up-to-date and grounded to factual and new information as it becomes available. As the current trend sees the size of these models to continually grow, mitigating those issues should rely on flexible and robust approaches that can be easily transferred to different domains and tasks.

Here, we aim to capitalize on the unique benefits offered by pre-trained LLMs and propose to overcome some of their limitations by drawing ideas from semi-parametric models (Khandelwal et al., 2019; Guu et al., 2020; Lewis et al., 2020; Izacard & Grave, 2020) that ground their decisions in external retrieved evidence to reduce hallucinations and improve factuality (Shuster et al., 2021). Specifically, we use the Internet as a source of up-to-date knowledge, and rely on the powerful few-shot capabilities of these LLMs to learn how to use it effectively for answering questions. Taking open-domain question answering as a task where factual correctness is vital, we design a system that given a question uses a retrieval model to retrieve relevant documents from the Internet. Then,

using few-shot learning we prompt the model to answer the question via conditioning on the retrieved documents, without the need to fine-tune or learn extra parameters.

As a retrieval system we use a search engine – in particular Google Search – allowing us to treat the whole web as a knowledge source. While Wikipedia has been the dominant knowledge source driving progress on a multitude of tasks, given the current progress and the quest towards more complex interactions, there has never been a better time to widen their scope, embracing the opportunities working with the whole web, such as considering a wider range of topics and views, as well as the many challenges, such as working with more noisy and potentially uncurated and unsafe text in the wild. Indeed, there is momentum building up in breaking away from Wikipedia-only research (Komeili et al., 2021; Nakano et al., 2021; Piktus et al., 2021; Thoppilan et al., 2022).

To test the effectiveness of equipping LLMs with Internet search on open-domain question answering, we use a mix of single-hop and multi-hop, language generation and classification tasks. We find that our biggest LLMs benefit from conditioning on the web through few-shot prompting. For the language generation tasks, we see a relative performance increase of 15%-30% over the commonly used closed-book few-shot approach, while also making up performance-wise for LLMs with smaller size. Surprisingly, we find that our method achieves gains, albeit smaller, even on complex multi-hop questions, despite the fact that these questions suffer from higher retrieval errors. While perhaps the mainstream view places scaling models' parameters as the primary way to increase their few-shot performance, our results add to the stream of work that emphasizes instead better use of the models' powerful prompting abilities (Rubin et al., 2021; Liu et al., 2021a). As such, our approach presents a lightweight method applicable to virtually any pre-trained LM without the need for fine-tuning or adding extra learnable parameters. Finally, increasing the *inference-time* compute of models via sampling multiple answers and reranking using scores computed from the same LLMs not only adds further performance gains, but also alleviates generally decreased performance of smaller few-shot LMs, partly closing their performance gap with larger models.

All in all, our findings hint at the possibility of slowing down the race towards the biggest model and instead shifting the attention to more targeted and effective use of models' few-shot capabilities in combination with increasing *inference-time* compute, a generally more scalable approach.

## 2 RELATED WORK

**Semi-parametric language models** have been recently gaining momentum (Khandelwal et al., 2019; Guu et al., 2020; Yogatama et al., 2021; Borgeaud et al., 2021), extending monolithic parametric models with information from a knowledge source. This process facilitates overcoming distribution shift (e.g., domain or temporal) in a flexible way by simply updating the external knowledge. When applied to question answering tasks (Lewis et al., 2020; Izacard & Grave, 2020; Sachan et al., 2021), they surpass performance of parametric-only models – they are able to efficiently handle an increasing number of retrieved passages and ground their predictions into additional information, thus reducing hallucinations and improving factuality. However, to be faithful to their input, these models need to be trained (or fine-tuned) to attend to the additional input. In contrast, our work pushes the limits of few-shot prompting as a way to learn to condition on external evidence with no additional training required, thus making it applicable to virtually any pre-trained LM.

**Web as knowledge source** Open-domain question answering traditionally has been studying carefully constructed benchmarks, where answerability of questions from Wikipedia has been confirmed through annotations. Recently a new trend emerged — using the whole web as knowledge source to support more varied and rich interactions. Augenstein et al. (2019) and Fan et al. (2020) make use of web data through commercial search engines as a part of building more diverse datasets for fact-checking. On the other hand, Piktus et al. (2021) find that considering the web as a retrieval source brings material gains to knowledge intensive tasks, despite any difficulties with building a search index from an order of magnitude more (noisy) data than Wikipedia. To avoid similar challenges with building and maintaining a search index, recent work that aims in improving factuality in user interactions adopts the use of commercial search engines as building block for their systems (Komeili et al., 2021; Nakano et al., 2021; Thoppilan et al., 2022; Menick et al., 2022). Similar to us, Nakano et al. (2021) analyze benefits of increasing compute at inference time. However, unlike us, they either target open-ended dialogue interactions (Komeili et al., 2021; Thoppilan et al., 2022) or focus on

optimizing performance using more intensive techniques like fine-tuning and reinforcement learning (Nakano et al., 2021; Menick et al., 2022). In our work, we take a more light-weight approach without introducing learnable parameters. We push the limits of few-shot prompting and emphasize the need to establish strong baselines, aiming at gaining insights into the strengths and weaknesses of this generally applicable, due to its simplicity, approach.

# 3 FEW-SHOT PROMPTING FOR INTERNET-AUGMENTED LANGUAGE MODELS

In this section, we describe our approach for improving the performance of pre-trained LMs in the task of open-domain question answering. Specifically, we propose to use few-shot prompting as a flexible and robust way to condition any pre-trained LSLM on external evidence, allowing for better grounding to factual and up-to-date information. Our approach consists of 3 steps (see Appendix A.1 for an illustration of the method). First, given a question we *retrieve* a set of relevant documents from the web using a search engine (§3.1). We then use the retrieved evidence to condition the LM through few-shot *prompting* (§3.2). Finally, we generate multiple candidate answers from each evidence and, to select the best answer, *rerank* them using scores computed using the same LM (§3.3).

## 3.1 RETRIEVE: GOOGLE SEARCH FOR DOCUMENT RETRIEVAL

For a question $q$, we need a set of relevant documents $D$ which would allow us to extend the model's knowledge to factual and (potentially) new information not already present in its weights. Towards more realistic and open-ended user interactions, we retrieve documents using an off-the-shelf search engine, i.e., Google Search. Specifically, we use each question $q$ verbatim as a query and issue a call to Google Search via the Google Search API.[1] For each question, we retrieve the top-20 urls and parse their HTML content[2] to extract clean text, resulting in a set of documents $D$ per question $q$.[3]

As documents in $D$ can originate from news or Wikipedia articles to whole books, they tend to be long, with an average length of 2,056 words. As this exceeds the input sequence length of models, we condition the model on shorter excerpts extracted from the documents in $D$. Specifically, we first chunk all documents into paragraphs of 6 sentences. We then embed $q$ and the paragraphs using TF-IDF and using cosine similarity we produce a (ranked) list of evidence paragraphs $P$, thus only using in the prompt smaller, more relevant parts of the full documents.

Overall, using Google Search allows us to tap into the diverse and ever-growing content on the web, instead of being confined on the information found in the static snapshots of the curated content on Wikipedia, a commonly used knowledge source in question answering tasks. In addition, we found that Google Search and the web offers also practical performance gains and is on-par (even sometimes outperforming) current Wikipedia-based state-of-the art dense retrievers (see Section 5 and comparisons on the BEIR benchmark in Appendix A.3). We discuss broader limitations and opportunities of using search engines in Section 6.

## 3.2 PROMPT: FEW-SHOT PROMPTING FOR CONDITIONING ON EVIDENCE

Given a question $q$ and a set of retrieved paragraphs $P$, we use few-shot prompting to condition pre-trained LMs on the paragraphs. This is done by taking the conventional $k$-shot prompting for (closed-book) QA, that only considers tuples of ⟨questions, answers⟩, and extending it with an evidence paragraph, resulting in a prompt of the form

```
Evidence: ...
Question: ...
Answer: ...
```

In all experiments we set $k = 15$. In Section 4 we give details on how we obtain the ⟨evidence, question, answer⟩ triplets to populate the prompt.

---

[1]https://developers.google.com/custom-search. Collecting evidence for 20k questions would incur costs of a few hundred dollars.

[2]We access and use - for research and publication purposes - a search corpus comprising text taken from the open web, created in accordance with generally agreed standards for inclusion/exclusion of content.

[3]To prevent data leakage, we filter out urls of the official dataset websites from the returned urls.

While experimenting with prompts, we found that swapping the question with the evidence, thus increasing the distance between questions and answers, yielded consistently lower results across all our datasets. We believe this is another manifestation of LMs' struggles to incorporate information from longer contexts; further performance increase could be achieved by selecting in-context examples for the prompt retrieved based on similarity to the question being answered (Liu et al., 2021b).

### 3.3 RERANK: INCREASING INFERENCE-TIME COMPUTE VIA ANSWER RERANKING

Increasing compute of models via scaling the number of parameters, hence increasing *training-time* compute, has been shown to improve performance of models in various few-shot tasks (Brown et al., 2020; Rae et al., 2021). Here, we ask whether similar gains can be obtained when scaling the *inference-time* compute. through sampling multiple answers from the model, which we then rerank using different probabilistic factorizations of the question answering task as scoring functions.

Specifically, as $P$ contains retrieved paragraphs ordered via their similarity to question $q$, we select the top $n = 50$, use each one separately to prompt the model and produce multiple candidate answers for each paragraph (for classification tasks we produce a distribution over all class labels for each paragraph). Overall, this allows us to consider a larger number of possible answers candidates while overcoming potential retrieval errors – considering more paragraphs increases retrieval recall.

Given an answer $a_i$ for a question $q$ conditioned on each of the $n$ retrieved paragraph $p_i$, we consider the following ways for estimating the answer probability: (i) direct inference, picks an answer that maximizes $p(a \mid q) = \sum_{i=1}^{n} p_{tfidf}(p_i \mid q) . p(a_i \mid q, p_i)$, referred to as RAG here and inspired from the model of Lewis et al. (2020) (ii) Noisy channel inference, picks an answer that maximizes $p(a_i, q \mid p_i) = \frac{p(q \mid a_i, p_i) \cdot p(a_i \mid p_i)}{p(q \mid p_i)}$, (Echihabi & Marcu, 2003; Lewis & Fan, 2019) (iii) Product-of-Experts (PoE), which combines all probabilities used above, in addition to $p(p_i \mid q)$.[4]

**Pre-trained LSLMs as scorers** All conditional probabilities are computed using LMs that we $k$-shot prompt for producing the respective distributions ($k = 15$) (see Appendix A.7 for example prompts). In this way, we turn LMs into models of arbitrary probabilities, which to the best of our knowledge is something not explored before. Exception to this is $p_{tfidf}(p_i \mid q)$ that is computed as the normalized cosine similarities between the TF-IDF passage and question representations – as passages can be long, getting reliable estimates of $p(p_i \mid q)$ using prompted LMs was challenging.

## 4 EXPERIMENTAL SETUP

**Datasets** We use 3 question answering datasets: the single-hop NQ (Kwiatkowski et al., 2019), and the multi-hop HOTPOTQA (Yang et al., 2018) and STRATEGYQA (Geva et al., 2021); and a fact-checking multi-hop dataset FEVER (Thorne et al., 2018). We select these datasets as they allow us to consider a mixture of language generation (NQ, HOTPOTQA) and classification (2-way for STRATEGYQA, 3-way for FEVER) tasks, as well as single- and multi-hop questions. We represent all items in the datasets as tuples $\langle q, A, G \rangle$, where $q$ is the question, $A$ is the the set of possible answers (or a single class label for the classification datasets), and $G$ is the set of gold evidence documents provided by the dataset. For the few-shot learning, we use the prompt format presented in Section 3.2 and create dataset-specific 15-shot prompts (for each of the datasets, we use the same set of questions for the few-shot examples), for a total of 4 prompts (for computing different scoring probabilities), populating them with the necessary $\langle$evidence, question, answer$\rangle$ triplets from each dataset.[5] As evidence, we use the gold evidence documents in $G$.

**Evaluation metrics** To evaluate the performance on these tasks, we report exact match for generation tasks and accuracy for classification tasks. Moreover, to better understand the interaction between retrieval and subsequent QA performance, we introduce a retrieval score. For generation tasks, we calculate answer recall in the conditioning evidence paragraphs $P$. For classification tasks, since the answer takes the form of a class label, we instead compute the normalized word overlap (excluding stopwords) between the gold paragraph $G$ and each of the conditioning paragraphs in $P$, and report the maximum word overlap achieved among the paragraphs $P$.

---

[4]The interpolation weights of probabilities are optimized on a held-out set of 10% of data.

[5]See Appendix A.6 for the prompts we used for each dataset.

**Language Models**  All experiments in this work use the GOPHER LM of 280 billion parameters (Rae et al., 2021). Alongside this model, and in order and to answer questions regarding scaling, we also use smaller models of the same family abbreviated by their number of parameters, i.e., 44M, 117M, 400M, 1B and 7B. Besides the number of parameters, the models also differ in the input sequence length: *2048* for 7B and 280B, and *1024* for 44M, 117M, 400M, and 1B. All models were trained on MassiveText for 300 billion tokens. For generation tasks, we use nucleus decoding, with parameters 0.8 for the probability cut-off and temperature of 1.

**Open-domain question answering models**  Here, we describe the open-domain question answering systems we build, all based on LMs presented above. We first describe our open-book models (referred to as OB), which condition on the provided evidence to generate an answer. $OB_{Google}$ will use the Google retrieved paragraphs. For each question $q$ we will generate 4 candidate answers $a'$ for each of the 50 paragraphs $p$, for a total of 200 answers. We will then select the best answer $a^*$ as $\arg\max_{a'} f(a', p, q)$, where $f$ is a pre-defined scoring function. In its basic form, $OB_{Google}^{a|q,p}$ will use as scoring function $f$ only the probability returned by the question answering model. Models $OB_{Google}^{\text{Noisy Channel}}$, $OB_{Google}^{\text{RAG}}$ and $OB_{Google}^{\text{PoE}}$ will use the noisy channel, direct inference and PoE factorizations, introduced in Section 3.3. Moreover, to better assess the performance of few-shot prompting, we design a model that assumes an oracle retrieval system – $OB_{Gold}$ conditions on gold evidence passages for each question provided by the datasets.

Finally, as a baseline model, we will use a pre-trained LM *without* evidence, prompting it with ⟨question, answer⟩ tuples. This is the conventional way found in the literature of few-shot language models (Brown et al., 2020) for performing open-domain question answering. These models are usually referred to as *closed-book*, as they do not use any knowledge source (unlike open-book), but solely rely on the knowledge encoded in their weights. We refer to these models as CB.

To fairly compare the different models, we need to account for the fact that $OB_{Google}$ searches for the right answer in a big pool of candidates generated by conditioning on multiple paragraphs; for both CB and $OB_{Gold}$ we sample 200 answers, and select the one with the highest probability. [6]

## 5  RESULTS

### 5.1  CONDITIONING A LARGE-SCALE LANGUAGE MODEL ON GOOGLE SEARCH RESULTS

| Dataset | SOTA | #sampled answers: 1 | | #sampled answers: 200 | | | | Retrieval performance@50 |
| | | CB | $OB_{Google}^{\text{no reranking}}$ | CB | $OB_{Gold}$ | $OB_{Google}^{a|q,p}$ | $OB_{Google}^{\text{PoE}}$ | |
|---|---|---|---|---|---|---|---|---|
| NQ | 55.9[a] | 21.7 | 23.1 | 25.8 | 61.7 | 32.7 | 38.4 | 85.0 |
| HOTPOTQA | 65.2[b] | 20.7 | 24.5 | 21.2 | 54.8 | 26.3 | 30.3 | 55.5 |
| FEVER | 73.2[c] | 44.5 | 52.2 | 44.5 | 66.6[e] | 52.0 | 57.2 | 43.3 |
| STRATEGYQA | 63.6[d] | 61.0 | 61.1 | 61.0 | 80.4 | 64.6 | 66.2 | 34.9 |

[a] Fajcik et al. (2021) [b] Yang et al. (2018) [c] Kruengkrai et al. (2021) [d] Geva et al. (2021) [e] Note that in this work we use probabilities of class labels directly, whereas in Rae et al. (2021) these probabilities were used as features for a classification model based on multi-class logistic regression.

Table 1: Results on 4 question answering datasets using the GOPHER-280B model.

We assess the effectiveness of our method for improving the performance of pre-trained LMs on open-domain question answering. We start with Gopher-280B, the best LM among the ones considered in this work; we test whether using few-shot prompting as a way to condition on external retrieved evidence (i.e., $OB_{Google}^{a|q,p}$) improves the performance of Gopher-280B over its closed-book version (i.e., CB), which uses knowledge solely in its weights. Table 1 presents the results on our 4 datasets.

Overall, we find that, indeed, conditioning Gopher-280B on the Google Search results leads to improved performance on all datasets, despite the fact that both Gopher-280B open- and closed-models use the same underlying LM. We see stronger performance over the closed-book version for the generation tasks, with the relative improvements reaching 30% on the NQ dataset. For NQ and HOTPOTQA, stronger performance is driven by higher retrieval recall and strong extractive

---

[6]Performance plateaued at 50 samples – sampling many answers with the same prompt decreases diversity.

behaviour of the model, very frequently producing an answer present in the conditioning evidence, a welcomed feature for extractive tasks (i.e., generated answer is present in evidence in 89.4% and 70.5% cases, for NQ and HOTPOTQA respectively). We also see gains, albeit smaller in scale, for the more reasoning classification tasks, indicating that few-shot prompting with retrieved evidence is beneficial over and above its extractive power. Appendix A.8 presents a manual error analysis.

**Retrieval performance**   Turning to the retrieval performance achieved by Google Search (see *Retrieval performance* column in Table 1), we see that the effectiveness of our approach of using the question verbatim as a search query heavily depends on the complexity of the questions in each dataset. Compare for example a typical question in the single-hop NQ "How many episodes in season 2 breaking bad?", where recall@50 reaches 85%, with one from HOTPOTQA "What is the relationship of Yeshahework Yilma's mother to the man who was Ethiopia's emperor from 1930 to 1974", where recall performance drops to 55.5%. Indeed, performance on multi-hop datasets (HOTPOTQA and STRATEGYQA) is generally worse than in the single-hop ones (NQ and FEVER). In fact, STRATEGYQA sees the smallest relative increase in performance with respect to the closed-book version. Questions in this dataset, albeit somewhat artificial, are inventive and involve linking non-trivial, normally unrelated facts (e.g., "Could a sloth hypothetically watch an entire episode of Scrubs underwater?"), stress-testing performance of retrieval models and calling for more sophisticated *learning to search* approaches (Adolphs et al., 2021; Komeili et al., 2021; Nakano et al., 2021).

Compared to current Wikipedia-based state-of-the-art models, which require training on the specific datasets, we find that the generic Google Search retrieval outperforms DPR (Karpukhin et al., 2020) on NQ (DPR recall@50 84%), while is only marginally outperformed by MDR (Xiong et al., 2020) on HOTPOTQA (MDR recall@20 52.1% vs ours recall@20 50.1%). Google Search acts here as a zero-shot retrieval, capable of generalizing on different tasks and knowledge corpus. Finally, our probabilistic reranking further improves performance: across all datasets, $OB_{Google}^{PoE}$ outperforms consistently the simpler $OB_{Google}^{a|q,p}$, widening the gap between closed-book and Google-conditioned model. We provide more in-depth discussion of reranking results in Section 5.2.

## 5.2   ABLATIONS

**Effect of Reranking**   In Table 1 (see columns *#sampled answers:1*), we ablate the use of ranking – even conditioning on a single Google evidence can bring gains over the closed-book model, evident by the higher performance achieved by $OB_{Google}^{no\ reranking}$ compared to CB.

**Effect of different scoring functions**   Figure 1 presents a comparison of the 4 different scoring functions introduced in Section 3.3. Note that, besides $p(a|q,p)$, which is computed using Gopher-280B, the remaining probabilities used to compute the 3 scores, i.e., PoE, RAG and Noisy Channel, are obtained by few-shot prompting the smaller 7B model. We find that that reranking with factorizations that consider scores beyond just the answer probability improve performance across all datasets, despite the fact that we do not train specialized models for deriving the extra scores but rather use an order of magnitude smaller prompted LMs.

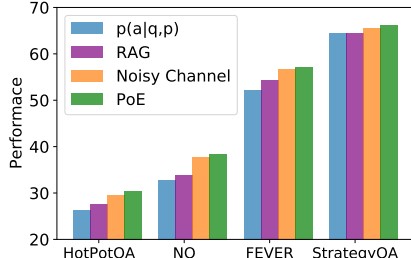

Figure 1: Question answering performance on all 4 datasets using Gopher-280B $OB_{Google}$ model in combination with different answer scoring functions for reranking the answers generated for each of the top 50 Google retrieved paragraphs.

Looking at the individual scores, we find $p(a\,|\,q,p_i)$ and $p(q\,|\,p_i,a)$ to be most informative – this last probability captures how well the model can "explain" the question given a retrieved evidence and an answer. In fact, as also show in Appendix A.2, this score could be reliably used for reranking passages that are more predictive of the final answer. On the other hand, $p(p_i\,|\,q,a)$, while being correlated to $p(q\,|\,p_i,a)$, has a higher variance (likely due to varying length of the passages) and is much less informative overall. Among the three factorizations that we consider, the lowest performance is consistently observed for the RAG-inspired factorization. We attribute this to the way the passage relevance is derived: as we do not use $p(p_i\,|\,q,a)$ from the model, we instead rely

on normalized cosine similarities between TF-IDF vectors which, as we show in Appendix A.2, are more reliable than the passage probabilities from the LM, but less accurate than $p(q \mid p_i, a)$.

While more elaborate scores are able to reduce the gap with the in-domain fine-tuned models (compare column *SOTA* with column $\text{OB}_{\text{Google}}^{\text{PoE}}$ in Table 1), in line with previous work (Wei et al., 2021) we find that few-shot prompting still generally lags behind the specialist models. However, this should be considered in perspective of how generalizeable our setup is to a *diverse* set of questions, from single-hop to multi-hop to fact-checking, turning any LM to a retrieval-augmented LM.

**Oracle retrieval**    The results of $\text{OB}_{\text{Gold}}$ using gold evidence, hence assuming oracle retrieval, can be treated as upper-bound. Results suggest that there is room for improvement in relying more on the conditioning evidence. We envision several directions: if we remain within the few-shot paradigm, more accurate prompt optimization, like in-context example selection, can further boost results (Liu et al., 2021b), while, constrained decoding (Post & Vilar, 2018; Lu et al., 2021) can be a way to condition the model at inference-time, by explicitly grounding an answer in the provided evidence.

## 5.3    SCALING ANALYSIS OF OPEN- AND CLOSED-BOOK MODELS

So far, we observed that conditioning a 280 billion LM on retrieved evidence from Google Search resulted in improved performance over the closed-book version of the same LM. But are these gains only confined on the (very) large-scale regime or could this method be used to also boost performance of smaller models? To answer this, we compute the open- and closed-book performance of 5 additional models containing 44M, 117M, 400M, 1B and 7B parameters. We follow the same approach as before; we use $k$-shot learning to prompt for the open-book models using Google Search results and $k$-shot learning to prompt for ⟨question, answer⟩ for the closed-book models, with k = 15. In Figure 2 we present results as a function of models' parameters in millions, for open-book (in solid lines) and closed-book models (dashed lines) models. We report accuracy for the classification tasks (left figure) and exact match for the generation tasks (right figure).

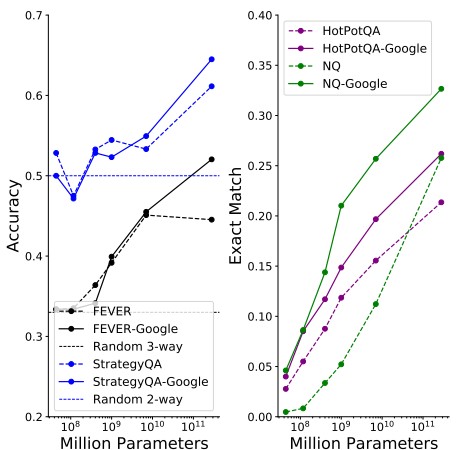

Figure 2: Scaling analysis of open-book (solid lines) and closed-book (dashed lines) question answering models using LMs of varying sizes ranging from 44M to 280B parameters.

Conditioning smaller models on retrieved evidence is particularly beneficial for the generation tasks, which given reliable evidence can be more extractive in nature. In fact, making use of external evidence does not simply improve the factuality of the models through grounding to external sources, but in many cases results in smaller open-book models outperforming bigger closed-book models: for both NQ (in green lines) and HOTPOTQA (in purple lines) the open-book 7B model is on par with the several times larger closed-book 280B model and for NQ the 1B is only 4% worse than the the 200x larger closed-book 280B model. Despite the simplicity of the approach and the potential errors introduced by the retrieval method, conditioning through few-shot prompting can, to some extent, make up for a smaller model size.

However, we also see that for the more reasoning tasks, i.e., FEVER (in black lines) and STRATEGYQA (in blue lines), the gains manifest mostly in the 7B models, though are smaller. For the (relatively) smaller models, their closed-book performance fluctuates around the random baseline. As such, grounding to factual information is perhaps a secondary concern in those cases; retrieval can boost reasoning capabilities for models with pre-existing good reasoning skills (as in the case of 280B), but cannot compensate for general lack of reasoning abilities.

## 5.4 INCREASING INFERENCE-TIME COMPUTE

The main driver of performance improvements of few-shot LMs has been scaling their model size. Increasing training-time compute is thus spent in injecting (and potentially memorizing) Internet-scale data. Here, we put forward a different hypothesis; we posit that we can improve performance of smaller LMs by giving them direct access to the Internet, thus freeing *training-time* compute which could then be spent to increasing their *inference-time* compute.[7] While there are many ways to achieve that, here we choose to increase models' compute via using multiple Google retrieved paragraphs to sample candidate answers for each and then reranking those answers using the functions introduced in Section 3.3. We focus on the 3 biggest models used in this study, comparing the open-book version of 1B and 7B with the closed-book version of Gopher-280B. We conduct this study on NQ.

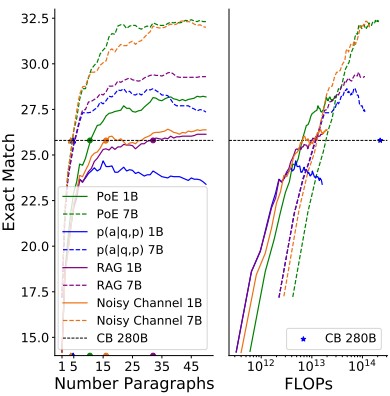

Figure 3-left presents exact match performance on NQ as a function of number of paragraphs (and hence candidate answers we are sampling and reranking). We see that considering more than the top-1 paragraph improves performance of models. Moreover, as evident by the slight upward slops, considering gradually more paragraphs benefits those scoring functions that incorporate some notion of paragraph "goodness", i.e., RAG (in purple lines) and PoE (in green lines) and Noisy Channel (in orange lines). In contrary, reranking only using the probability $p(a|q, p)$ of the answer under the model (in blue lines) results in decreased performance after a certain point. This is potentially due to some pathological behaviour of the model, e.g., assigning high probability to candidate answers that exist in the conditioning evidence but are otherwise wrong. This is consistent with findings in Machine Translation which find that reranking only using the forward probability gives worse results as the number of candidates increases beyond a certain threshold (Stahlberg & Byrne, 2019).

Figure 3: Scaling analysis on the NQ dataset: exact match as a function of number of conditioned paragraphs (left) and FLOPs (right).

More interestingly, we find that using as little as 5 paragraphs from Google Search, in conjunction with the 7B model (in dashed lines for the different scoring functions) surpasses the performance of closed-book Gopher-280B model (horizontal black line), suggesting that searching the Internet is worth more than 273 billion parameters. Finally, Figure 3-right presents a similar plot, but as a function of FLOPs; this accounts for additional factors, on top of the number of Google retrieved paragraphs, like models' size and compute spent to calculate the scores for each of the scoring functions. Closed-book Gopher-280B spends compute for *implicitly* combining retrieving the facts that are memorized in its weights, and then reason over them. By *explicitly* outsourcing the memorization of facts and their retrieval to Google Search, the same *inference-time* compute can now be spent more effectively in reranking more samples with a smaller model. As such, for the same or less inference-time compute, reranking models achieve better performance, as indicated by all the lines being to the left of the blue dot in the horizontal dashed line, i.e., the FLOPs of the closed-book Gopher-280B.

## 5.5 KEEPING QA MODEL UP-TO-DATE

We now test whether using a commercial engine as a source of up-to-date information about the world can help stale models answer questions about new events. Since Gopher-280B was trained with data up to (and including) November 2020, questions about facts beyond that date would not have been seen in its training data. To derive such questions, we use the SituatedQA dataset (Zhang & Choi, 2021) which contains questions grounded in different dates. Table 2 presents the exact match results on the complete development set of

|  | Questions | |
|---|---|---|
|  | All | Post-2020 |
| CB | 26.3 | 15.8 |
| $OB^{a|q,p}_{Google}$ | 28.1 | 22.4 |

Table 2: Exact Match performance on SituatedQA.

---

[7]Khandelwal et al. (2019) also find that training smaller models with large datastores surpasses perplexity performance of bigger language models.

questions (*all*) – we also create a smaller subset of 80 questions about facts in 2021 and beyond to test adaptation to new information (*post-2020*).

The higher performance of $\text{OB}_{\text{Google}}^{\text{a}|\text{q,p}}$ indicates that using few-shot prompting as a way to condition on evidence is an effective way of incorporating truly new information into the QA system. However, despite having access to updated information, the performance of $\text{OB}_{\text{Google}}^{\text{a}|\text{q,p}}$ on *post-2020* questions is substantially lower than the performance on the complete set of questions, suggesting that conflicting parametric (i.e., in language model) and contextual (i.e., in retrieved evidence) knowledge poses challenges for retrieval-based models (Longpre et al., 2021).[8]

## 5.6 REPRODUCIBILITY

We take steps to reproduce the proposed technique using publicly available material; Appendix A.4 presents results using our prompts and data in conjunction with the OPT-6B model (Zhang et al., 2022) and Appendix A.5 presents results using our language model and prompts in conjunction to Wikipedia-based DPR evidence (Izacard & Grave, 2020).

## 6 DISCUSSION

Towards more open-ended and "in the wild" user interactions, in this work we presented a straightforward method targeted to alleviate some of the challenges faced by LSLMs with respect to grounding to factual and up-to-date information. The core of our method lies in combining the powerful few-shot abilities of LSLMs with a state-of-the-art retrieval model, i.e., Google Search, for access to a broad and constantly updated knowledge source, the whole web. We applied our method on open-domain question answering and found that, despite its simplicity, using few-shot prompting to condition models on the web provides an effective approach for increasing models' factuality. Improvements were not just confined to the largest LMs; we saw increases in performance across the board of model sizes, with smaller open-book models often surpassing performance of bigger few-shot closed-book models. Further gains were achieved when increasing inference-time compute via using multiple retrieved evidences to generate multiple answers followed by a reranking stage that uses scores computed by the same LMs. Our approach offers a simple way to turn virtually any pre-trained LM to a retrieval-augmented LM model without the need for fine-tuning or adding learnable parameters.

Mainstream view places scaling LMs' parameters as the main way to improve their few-shot performance. However, significant gains can be achieved by inference-type interventions, i.e., more targeted use of few-shot abilities and increase of inference-time compute. This may slow down the race towards the biggest LM, shifting attention towards finding more effective ways to use models.

**Limitations** Despite their progress, LLMs are still sometimes outperformed by fine-tuned, and even smaller, models (Wei et al., 2021). While our targeted interventions were successful in closing this gap on open-domain question answering, we are still behind in-domain fine-tuned models. For reasoning tasks, retrieval was able to improve only the largest amongst the considered models. Moreover, while we have considered a wide-range of diverse question answering datasets, our current experiments only capture a small fraction of (simple) user interactions where factuality plays a crucial role.

Finally, in an attempt to work towards open-ended interactions and improve our system's performance, we used a commercial search engine as a retrieval model, allowing us to work with the whole web as a knowledge source. Since we are not confined to working only with the curated and "sanitized" Wikipedia, we anticipate a number of safety issues to arise, including misinformation, harmful content. While we have relied on the underlying safety nets of the search engines we use, more work should be put in place scoping out and better understanding the risks and, most importantly, providing effective ways to mitigate those. As working with the whole Web gains more momentum, we expect to see more work that surfaces and tackles these points. Another potential concern is reproducibility of the research results given that we do not have as tight control over retrieval results as in the case of offline retrieval.This might, indeed, create some discrepancies over longer time horizons, not only due to changes in the underlying search engine logic, but also because new published documents might provide more direct answers. Overall we believe that potential benefits of understanding how to use Google Search with LLMs overweight the potential downsides, if done responsibly.

---

[8]Published test set results for fine-tuned open- and closed-book are 23.0% and 18.3% (Zhang & Choi, 2021).

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

# A APPENDIX

## A.1 ILLUSTRATION OF METHOD

Figure 4 presents an illustration of our method.

## A.2 USING LLM-DERIVED SCORES FOR PASSAGE RERANKING

Figure 5 presents results when using LLMs-derived scores to rerank passages.

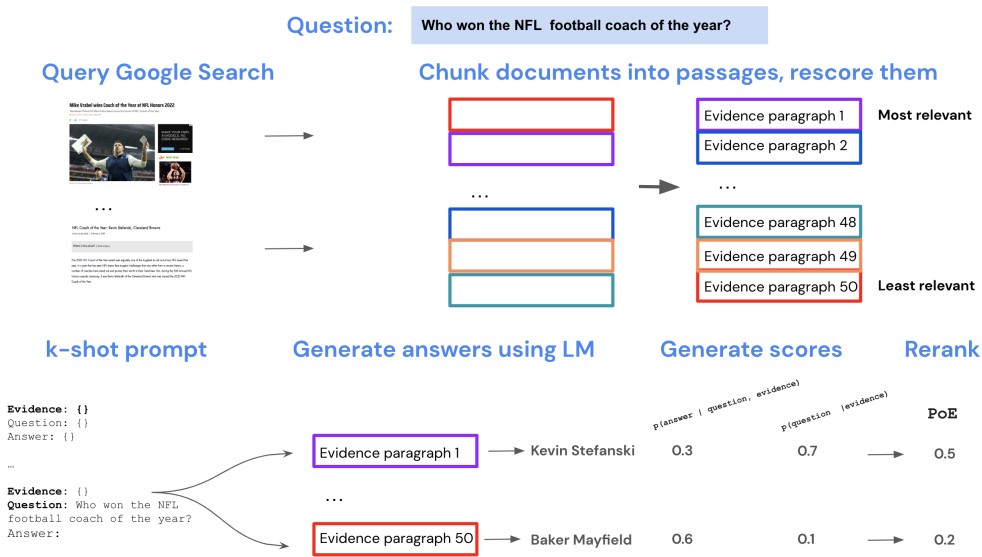

Figure 4: Schematic representation of the method presented in Section 3.

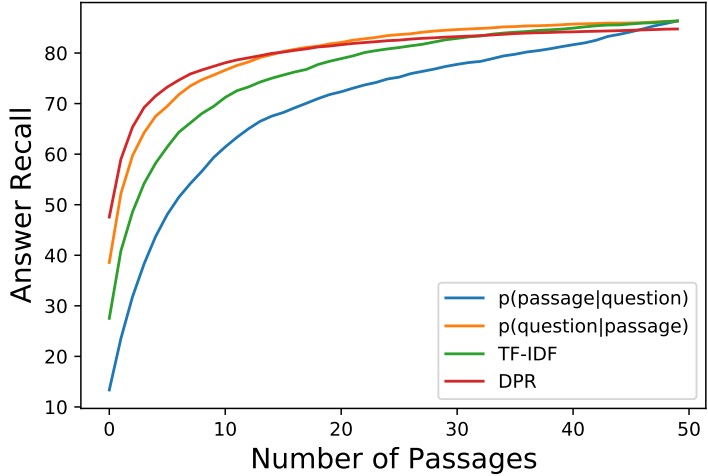

Figure 5: Comparing NQ answer recall when reranking Google Search passages using LLMs-derived scores against DPR retriever on Wikipedia.

### A.3 RETRIEVAL PERFORMANCE ON A SUBSET OF BEIR BENCHMARK

Over and above access to updated information, search engines are also optimized for good retrieval performance across a variety of sources, domains and queries. The discourse in the community is slowly moving away from in-domain performance of retrievers when trained with abundance of in-domain data, towards more realistic scenarios of out-of-distribution generalization using zero-shot evaluations. The recent BEIR benchmark (Thakur et al., 2021) provides such an evaluation and so we used this to compare the zero-shot retrieval performance of Google against DPR (we use the DPR model used in the BEIR benchmark pre-trained on NQ, TriviaQA, WebQuestions, and CuratedTREC). As the search spaces between DPR and Google are different (Google uses the whole web, DPR is using the search space provided for each domain by BEIR), we evaluated models using a word overlap metric between the gold and retrieved passages. We only report metrics for the subsets of BEIR that contain no more than 500k documents as search space.

|  | average | scidocs | scifact | nfcorpus | arguana | fiqa | trec_covid | webis_touche2020 |
|---|---|---|---|---|---|---|---|---|
| DPR@1 | 21.1 | 33.05 | 15.99 | 31.54 | 31.54 | 21.91 | 16.44 | 10.59 |
| Google@1 | 43.7 | 56.85 | 32.48 | 60.59 | 60.59 | 57.07 | 37.37 | 35.50 |
| DPR@100 | 50.3 | 33.66 | 78.44 | 28.79 | 85.57 | 55.41 | 24.14 | 46.44 |
| Google@100 | 67.7 | 68.67 | 76.00 | 55.17 | 86.18 | 78.97 | 54.03 | 55.16 |

Table 3: Word overlap results between retrieved and gold evidence on subsets of the BEIR benchmark for different retrieval models.

Looking into the domain-averages performance in Table 3, we see that Google@1 is +22% better than DPR across all domains considered here. We note that these domains do not include the Wikipedia based ones (e.g., NQ and HotpotQA) in which we know DPR scores high. On looking at top100, we see that for certain domains (e.g., nfcorpus, arguana) DPR approaches Google. The difference between Google and DPR for top100 is smaller than top1, but DPR is still outperformed by google by 17%.

## A.4 Reproducing results with OPT-6B model

We performed an experiment using the OPT model of 6.7 billion parameters. We measured the performance of OPT-CB and OPT-OB, following the same experimental setup we followed for the experiments using Gopher-280Bi.e., same number of evidences used, same number of samples taken and the same prompts). We found the performance of the OTB-CB to be 15% (compared the similar-sized 7B-CB 11% from Figure 2) and the OTB-OB to be 24% (compared to our 7B-OB 26% from Figure 2).

 of out-of-distribution generalization using zero-shot evaluations. The recent BEIR benchmark (Thakur et al., 2021) provides such an evaluation and so we used this to compare the zero-shot retrieval performance of Google against DPR (we use the DPR model used in the BEIR benchmark pre-trained on NQ, TriviaQA, WebQuestions, and CuratedTREC). As the search spaces between DPR and Google are different (Google uses the whole web, DPR is using the search space provided for each domain by BEIR), we evaluated models using a word overlap metric between the gold and retrieved passages. We only report metrics for the subsets of BEIR that contain no more than 500k documents as search space.

## A.5 Reproducing results using Wikipedia-based DPR evidence

We find that DPR+Gopher performance (32.9%) is almost indistinguishable from the Google+Gopher (32.7%), which is expected as the retrieval performance for both is almost the same. Since NQ has been created using Wikipedia articles, DPR has been trained on Wikipedia and the search space is the exact Wikipedia snapshot that was used to create the dataset, this experiment is the best possible case scenario for DPR. For example, we see that many questions in NQ are time-sensitive (e.g., When does thomas rhett's new album come out?) and so Google, which is using an updated snapshot of the world, is at disadvantage when compared to DPR with. The recent Atlas work (Izacard et al., 2022) report similar findings with respect to the effects of temporal alignment between the Wikipedia dumps used to create NQ and retrieval evidence.

## A.6 QA Prompts

Bellow we provide the prompts used for each of the datasets to create the open-book models. The prompts for building the closed-book model is derived by omitting the *Evidence* part of the prompt.

## A.6.1  NQ

Evidence: Top 20 rankings as of 16 October 2017 Rank Change Team Points Germany 1631 Brazil 1619 Portugal 1446 Argentina 1445 5
↪  Belgium 1333 6 Poland 1323 7 France 1226 8 Spain 1218 9 Chile 1173 10 Peru 1160 11 Switzerland 1134 12 England 1116 13 Colombia
↪  1095 14 Wales 1072 15 Italy 1066 16 Mexico 1060 17 Uruguay 1034 18 Croatia 1013 19 7 Denmark 1001 20 9 Netherlands 931 * Change
↪  from 14 September 2017 Complete rankings at FIFA.com
Question: who has been ranked no. 1 in the latest football rankings announced by fifa
Answer: Germany

Evidence: "Your Love" is a song by the English rock band the Outfield, taken from their debut album Play Deep (1985). The song was
↪  penned by the band's guitarist John Spinks.
Question: who sings i just want to use your love tonight
Answer: English rock band the Outfield

Evidence: Principal photography began on May 20, 2016, in Welch, West Virginia.
Question: where was the movie the glass castle filmed
Answer: in Welch, West Virginia

Evidence: No. Name Field Affiliation Date of Appointment Date of Retirement Roopa Ganguly Art Bharatiya Janata Party 04-Oct-2016
↪  03-Oct-2022 Sambhaji Raje Social work Bharatiya Janata Party 07-Jun-2016 03-May-2022 Suresh Gopi Art Bharatiya Janata Party
↪  25-Apr-2016 24-Apr-2022 Subramanian Swamy Economics Bharatiya Janata Party 25-Apr-2016 24-Apr-2022 5 Narendra Jadhav Economics
↪  Nominated 25-Apr-2016 24-Apr-2022 6 Mary Kom Sport Nominated 25-Apr-2016 24-Apr-2022 7 Swapan Dasgupta Journalism Nominated
↪  25-Apr-2016 24-Apr-2022 8 K.T.S. Tulsi Law Nominated 25-Feb-2014 24-Feb-2020 9 K. Parasaran Law Nominated 09-Jun-2012
↪  28-Jun-2018 10 Rekha Art Nominated 27-Apr-2012 26-Apr-2018 11 Sachin Tendulkar Social service Nominated 27-Apr-2012 26-Apr-2018
↪  12 Anu Aga Business Nominated 27-Apr-2012 26-Apr-2018
Question: who was the first lady nominated member of the rajya sabha
Answer: Mary Kom

Evidence: The McChicken is a chicken sandwich sold by the international fast-food chain McDonald's. The sandwich consists of a
↪  toasted wheat bun, a breaded chicken patty, shredded lettuce, and mayonnaise.
Question: what is on a mcchicken sandwich from mcdonalds
Answer: a breaded chicken patty

Evidence: Life of Pi is a Canadian fantasy adventure novel by Yann Martel published in 2001. The protagonist is Piscine Molitor "Pi"
↪  Patel, an Indian boy from Pondicherry who explores issues of spirituality and practicality from an early age. He survives 227
↪  days after a shipwreck while stranded on a lifeboat in the Pacific Ocean with a Bengal tiger named Richard Parker.
Question: what is the tigers name in life of pi
Answer: Richard Parker

Evidence: Malware, short for malicious software, is an umbrella term used to refer to a variety of forms of harmful or intrusive
↪  software, including computer viruses, worms, Trojan horses, ransomware, spyware, adware, scareware, and other malicious programs.
↪  It can take the form of executable code, scripts, active content, and other software. Malware is defined by its malicious
↪  intent, acting against the requirements of the computer user -- and so does not include software that causes unintentional harm
↪  due to some deficiency.
Question: the general term for software that is designed to damage disable or steal data is
Answer: Malware

Evidence: Mum Genre Sitcom Created by Stefan Golaszewski Written by Stefan Golaszewski Directed by Richard Laxton Stefan Golaszewski
↪  Starring Lesley Manville Peter Mullan Sam Swainsbury Lisa McGrillis Opening theme Cups by Lulu and the Lampshades Country of
↪  origin United Kingdom Original language (s) English No. of series No. of episodes 12 (to 27 March 2018) Production Running time
↪  30 minutes Production company (s) Big Talk Productions Distributor ITV Studios Release Original network BBC Two (2016-present)
↪  BBC Two HD (2016-present) Picture format 16: 9 1080i Audio format Stereo Original release 13 May 2016 (2016-05-13) -- present
Question: who sings the theme tune to mum on bbc2
Answer: Lulu and the Lampshades

Evidence: The Chess World Cup 2017 was a 128-player single-elimination chess tournament, held in Tbilisi, Georgia, from 2 to 27
↪  September 2017. It was won by Armenian grandmaster Levon Aronian. This was the second time he had won the Chess World Cup, 12
↪  years after his first win in 2005.
Question: where was the world chess tournament 2017 held
Answer: Tbilisi, Georgia

Evidence: T.J. Miller as Randy Kevin Michael Richardson as Rosie, others David Koechner as Robert "Bob Pogo" Pogrohvich, Frank's
↪  obese, chainsmoking boss. Kevin Farley as Babe, Carl, others Gary Cole as Rodger Dunbarton, the owner and founder of the
↪  airlines where Frank and his co-workers work. Joe Buck as Lou Gagliardi, others John DiMaggio as Scoop Dunbarton, Roger
↪  Dunbarton's racist and moronic nephew. Allison Janney as Henrietta Van Horne T.J. Miller as Randy Michael K. Williams as Smoky
Question: who voices randy in f is for family
Answer: T.J. Miller

Evidence: Las Vegas Stadium is the working name for a domed stadium under construction in Paradise, Nevada for the Las Vegas Raiders
↪   of the National Football League (NFL) and the UNLV Rebels football team from the University of Nevada, Las Vegas (UNLV). It is
↪   located on about 62 acres west of Mandalay Bay at Russell Road and Hacienda Avenue and between Polaris Avenue and Dean Martin
↪   Drive, just west of Interstate 15. Construction of the $1.9 billion stadium began in September 2017 and is expected to be
↪   completed in time for the 2020 NFL season.
Question: where are they building the new raiders stadium
Answer: Paradise, Nevada

Evidence: At the time of its initial public offering (IPO) on the stock market in June 1992, Starbucks had 140 outlets, with a
↪   revenue of US $ 73.5 million, up from US $1.3 million in 1987. The company's market value was US $271 million by this time. The
↪   12% portion of the company that was sold raised around US $25 million for the company, which facilitated a doubling of the
↪   number of stores over the next two years. By September 1992, Starbucks ' share price had risen by 70% to over 100 times the
↪   earnings per share of the previous year.
Question: when did starbucks become a publicly traded company
Answer: June 1992

Evidence: At the end of December 31, 2015, its employee strength was 170,664. Abid Ali Neemuchwala was appointed as Wipro's CEO
↪   after T.K. stepped down in early 2016.
Question: who become the ceo of it wipro company in 2016
Answer: Abid Ali Neemuchwala

Evidence: WINNER: Geoffrey Zakarian Episode 5 6 7 8 Comments Zakarian WIN IN IN CO IN CO WIN WIN The Next Iron Chef Falkner IN IN
↪   WIN IN CO WIN CO OUT Elim: Pressure Chiarello IN CO IN WIN IN IN OUT Elim: Passion Guarnaschelli IN WIN IN IN IN IN OUT Burrell
↪   IN IN IN IN WIN OUT Elim: Risk Samuelsson CO IN IN IN OUT Elim: Storytelling MacMillan WIN IN CO OUT Elim: Improvisation Hughes
↪   IN IN OUT Elim: Ingenuity Irvine IN OUT Elim: Transformation Mendelsohn OUT Elim: Resourcefulness
Question: who wins the next iron chef super chefs
Answer: Zakarian

Evidence: Support for and the elevation of leaves, flowers and fruits. The stems keep the leaves in the light and provide a place
↪   for the plant to keep its flowers and fruits. Transport of fluids between the roots and the shoots in the xylem and phloem
↪   Storage of nutrients Production of new living tissue. The normal lifespan of plant cells is one to three years. Stems have cells
↪   called meristems that annually generate new living tissue.
Question: what are the main functions of the stem
Answer: Production of new living tissue

## A.6.2 HotPotQA

Evidence: Seesaw is a musical with a book by Michael Bennett, music by Cy Coleman, and lyrics by Dorothy Fields. Michael Bennett
↪ (April 8, 1943 { July 2, 1987) was an American musical theatre director, writer, choreographer, and dancer. He won seven Tony
↪ Awards for his choreography and direction of Broadway shows and was nominated for an additional eleven.
Question: When was the writer of Seesaw born?
Answer: April 8, 1943

Evidence: Heinrich August Marschner (16 August 1795 { 14 December 1861) was the most important composer of German opera between
↪ Weber and Wagner. Carl Maria Friedrich Ernst von Weber (18 or 19 November 1786 5 June 1826) was a German composer, conductor,
↪ pianist, guitarist and critic, and was one of the first significant composers of the Romantic school.
Question: Heinrich Marschner was a composer who performed in the time frame after one of the first significant composers in what
↪ school of work?
Answer: Romantic

Evidence: The Elliott-Donaldson House is a historic mansion in Okolona, Mississippi, U.S.. It was built in 1850, a decade prior to
↪ the American Civil War of 1861-1865. By the end of the war, in 1865, Confederate States Army General Nathan Bedford Forrest
↪ stayed in the house to rest. It has been listed on the National Register of Historic Places since September 15, 1980. Nathan
↪ Bedford Forrest (July 13, 1821 { October 29, 1877), called Bedford Forrest in his lifetime, was a lieutenant general in the
↪ Confederate Army during the American Civil War.
Question: What lieutenant general stayed in the Elliott-Donaldson House?
Answer: Nathan Bedford Forrest

Evidence: Luca Parmitano (born 27 September 1976 in Paternò, Sicily) is an Italian engineer and astronaut in the European Astronaut
↪ Corps for the European Space Agency (ESA). The astronauts work on missions at the International Space Station. He was selected
↪ as an ESA astronaut in May 2009. Prof. Dr. Ulrich Hans Walter (born February 9, 1954) is a German physicist/engineer and a
↪ former DFVLR astronaut.
Question: who is younger Ulrich Walter or  Luca Parmitano?
Answer: Luca Parmitano

Evidence: A Boltzmann machine is a type of stochastic recurrent neural network (and Markov Random Field). Ludwig Eduard Boltzmann
↪ (February 20, 1844 { September 5, 1906) was an Austrian physicist and philosopher whose greatest achievement was in the
↪ development of statistical mechanics, which explains and predicts how the properties of atoms (such as mass, charge, and
↪ structure) determine the physical properties of matter (such as viscosity, thermal conductivity, and diffusion).
Question: What machine has the same name as another machine created by Ludwig Boltzmann?
Answer: Boltzmann machine

Evidence: Graham Linehan ( ; born 22 May 1968) is an Irish television comedy writer and director who, often in partnership with
↪ Arthur Mathews, has written or co-written a number of popular television comedies. He is most noted for the sitcoms "Father Ted",
↪ "Black Books" and "The IT Crowd". Amongst others, he has also worked on "Big Train", "Count Arthur Strong", "Brass Eye" and "The
↪ Fast Show". The IT Crowd is a British sitcom by Channel 4, written by Graham Linehan, produced by Ash Atalla and starring Chris
↪ O'Dowd, Richard Ayoade, Katherine Parkinson, and Matt Berry.
Question: Graham Linehan was the creator of the Ash Atalla-produced sitcom for what UK channel?
Answer: Channel 4

Evidence: Andrea Chénier is a verismo opera in four acts by the composer Umberto Giordano, set to an Italian libretto by Luigi
↪ Illica. It was first performed on 28 March 1896 at La Scala, Milan. The opera's story is based loosely on the life of the French
↪ poet André Chénier (1762{1794), who was executed during the French Revolution. The character Carlo Gérard is partly based on
↪ Jean-Lambert Tallien, a leading figure in the Revolution. "La mamma morta " (They killed my mother) is an aria from act 3 of the
↪ 1896 opera "Andrea Chénier" by Umberto Giordano.
Question: What is the name of the third act in a play where a character named Carlo Gérard is partly based on a revolutionary
↪ figure?
Answer: La mamma morta

Evidence: Richard Masur (born November 20, 1948) is an American actor who has appeared in more than 80 movies. From 1995 to 1999, he
↪ served two terms as president of the Screen Actors Guild (SAG). Masur currently sits on the Corporate Board of the Motion
↪ Picture & Television Fund. License to Drive is a 1988 teen adventure film written by Neil Tolkin and directed by Greg Beeman in
↪ his feature film directorial debut. It stars Corey Haim, Corey Feldman, Heather Graham, Carol Kane, Richard Masur, Michael
↪ Manasseri and Nina Siemaszko.
Question: License to Drive featured what future president of the Screen Actors Guild?
Answer: Richard Masur

Evidence: The British Broadcasting Corporation (BBC) is a British public service broadcaster with its headquarters at Broadcasting
↪ House in London. The BBC is the world's oldest national broadcasting organisation and the largest broadcaster in the world by
↪ number of employees. It employs over 20,950 staff in total, 16,672 of whom are in public sector broadcasting. The total number
↪ of staff is 35,402 when part-time, flexible, and fixed contract staff are included. Tenko was a television drama, co-produced by
↪ the BBC and the ABC.
Question: Where is the headquarters for the service broadcaster the show Tenko is on?
Answer: Broadcasting House in London

Evidence: Thomas Matthew "Tom" Chappell (born 1943) is an American businessman and manufacturer and co-founder of Tom's of Maine in
↪ 1970. Tom's of Maine is a brandname and manufacturer of natural-ingredients-only personal care products, a majority-owned
↪ subsidiary of Colgate-Palmolive since 2006. The company's products are intentionally mostly made without ingredients that are:
↪ chemically derived, have a negative environmental impact, or are tested on animals. While most of its products are vegan, some
↪ contain propolis and/or beeswax sourced from bees.
Question: Thomas Matthew "Tom" Chappell co-founded a commpany in 1970 that manufactures what products?
Answer: natural-ingredients-only personal care products

Evidence: Francis Bacon, 1st Viscount St Alban, {'1': ", '2': ", '3': ", '4': "} ( ; 22 January 15619 April 1626) was an English
↪ philosopher, statesman, scientist, jurist, orator, and author. He served both as Attorney General and as Lord Chancellor of
↪ England. After his death, he remained extremely influential through his works, especially as philosophical advocate and
↪ practitioner of the scientific method during the scientific revolution. James Spedding (28 June 1808 { 9 March 1881) was an
↪ English author, chiefly known as the editor of the works of Francis Bacon.
Question: James Spedding was chiefly known as the editor of the works of an author who served both as Attorney General and as what?
Answer: Lord Chancellor of England

Evidence: Romeo Montague (Italian: "Romeo Montecchi" ) is the protagonist of William Shakespeare's tragedy "Romeo and Juliet". The
↪   son of Montague and his wife, he secretly loves and marries Juliet, a member of the rival House of Capulet. Forced into exile
↪   after slaying Juliet's cousin, Tybalt, in a duel, Romeo commits suicide upon hearing falsely of Juliet's death. Benvolio is a
↪   fictional character in Shakespeare's drama "Romeo and Juliet". He is Montague's nephew and Romeo's cousin. Benvolio serves as an
↪   unsuccessful peacemaker in the play, attempting to prevent violence between the Capulet and Montague families.
Question: Which character does this protagonist, who secretly loves and marries a member of the rival house, of William
↪   Shakespeare's tragedy that has a fictional character Benvolio slay?
Answer: Tybalt

Evidence: Francesca Schiavone (] ; born 23 June 1980 in Milan) is an Italian tennis player who turned professional in 1998. She won
↪   the 2010 French Open singles title, becoming the first Italian woman to win a Grand Slam event in singles. She was also
↪   runner-up at the 2011 French Open. Her career high ranking is world No. 4, achieved on 31 January 2011. To date, Schiavone is
↪   the last one handed-backhand player to win a Grand Slam title on the women's tour. Carly Gullickson (born November 26, 1986) is
↪   a former American professional tennis player.
Question: What occupation have Carly Gullickson and Francesca Schiavone both held?
Answer: professional tennis player.

Evidence: Otello (] ) is an opera in four acts by Giuseppe Verdi to an Italian libretto by Arrigo Boito, based on Shakespeare's play
↪   "Othello". It was Verdi's penultimate opera, and was first performed at the Teatro alla Scala, Milan, on 5 February 1887. After
↪   Aida (original title: "Verdi's Messiah") is a 1985 play-with-music by Julian Mitchell. It is about Giuseppe Verdi, and the
↪   pressure put upon him after his attempt to retire from composing. Continued insistent prodding from his friends eventually
↪   results in one of his greatest masterpieces, the opera "Otello", which premiered in 1887.
Question: Where was the opera, which was the subject of After Aida, first performed?
Answer: Teatro alla Scala

Evidence: The Commodore 16 is a home computer made by Commodore International with a 6502-compatible 7501 or 8501 CPU, released in
↪   1984 and intended to be an entry-level computer to replace the VIC-20. A cost-reduced version, the Commodore 116, was sold only
↪   in Europe. In the middle of 1984 a Brazilian company called Prológica, which made its own versions of 8 bits US computers,
↪   brought to the Brazilian market a new equipment for its personal computer series called "CP" (shorten of Personal Computer in
↪   Portuguese).
Question: Were the Commodore 16 and Prológica CP-400 from the same country?
Answer: no

### A.6.3  STRATEGYQA

Evidence: The Albanian Declaration of Independence is written in Albanian, Gheg, Tosk, and Ottoman Turkish. The Arvanite Greek's are
↪  a major Tosk speaking group of southern Albania.
Question: Can an Arvanite Greek understand some of the Albanian Declaration of Independence?
Answer: true

Evidence: An anxious person may benefit from medication or therapy. The Wizard of Oz cannot give courage to anyone.
Question: Would an anxious person benefit from receiving courage from the Wizard of Oz?
Answer: false

Evidence: Silicon is a key material for the production of semiconductor chips. A silicon shortage would mean fewer semiconductor
↪  chips could be produced. A business that produces fewer products than normal will receive lower than normal revenue.
Question: Would a silicon shortage be bad for Intel's sales?
Answer: true

Evidence: The Superbowl is the championship game of the National Football League The National Football League is a sports league for
↪  American football American football enjoys the majority of its popularity in the United States The bengal fox is found
↪  exclusively on the Indian subcontinent
Question: Is a bengal fox likely to see the Superbowl?
Answer: false

Evidence: The letter B is the second letter in the Latin Alphabet. There was one total lunar eclipse in 2008.
Question: Does the letter B's place in alphabet exceed number of 2008 total lunar eclipses?
Answer: true

Evidence: The Battle of Baghdad was the U.S. invasion of Baghdad in the year 2003. Justin Bieber's album Believe was released in
↪  2012.
Question: Did U.S. soldiers listen to Justin Bieber's Believe album during the Battle of Baghdad?
Answer: false

Evidence: The Italian Renaissance was a period of history from the 13th century to 1600. A theocracy is a type of rule in which
↪  religious leaders have power. Friar Girolamo Savonarola was the ruler of Florence, after driving out the Medici family, from
↪  November 1494 { 23 May 1498.
Question: Was Florence a Theocracy during Italian Renaissance?
Answer: true

Evidence: The Tohoku earthquake led to the Fukushima Daiichi nuclear power plant meltdown Nuclear meltdowns lead to a release of
↪  deadly levels of radiation Godzilla draws power from radiation and is not hurt by it
Question: Could Godzilla have been killed by the Tohoku earthquake?
Answer: false

Evidence: Robert Moses Grove was a baseball player nicknamed Lefty Grove. Pablo Escobar had several nicknames including: Don Pablo,
↪  El Padrino, and El Patrón.
Question: Did Pablo Escobar's nickname collection outshine Robert Moses Grove's?
Answer: true

Evidence: Anaheim is the biggest city in Orange County, California Anaheim was founded by fifty German families People from Germany
↪  speak German
Question: Did the founders of the biggest city in Orange County, California speak Italian?
Answer: false

Evidence: Aerosmith is an American rock band that has five active members. The 2020 Mitsubishi Outlander has flexible seating that
↪  allows for seven seat capacity.
Question: Can Aerosmith fit in a 2020 Mitsubishi Outlander?
Answer: true

Evidence: The War in Vietnam (1945-46) lasted around 6 months. The gestation period for a llama is 11 months.
Question: Could a llama birth twice during War in Vietnam (1945-46)?
Answer: false

Evidence: Ivan the Terrible was nicknamed terrible because of his harsh rule. Ivan the Terrible's father, Vasili III Ivanovich, was
↪  nicknamed Vasili the Adequate. Ivan the Terrible's grandfather, Ivan III Vasilyevich, was nicknamed Ivan the Great.
Question: Did Ivan the Terrible's father and grandfather have nicer nicknames?
Answer: true

Evidence: The Beatles were active from 1960 until 1969. Disco began to appear around 1972.
Question: Did the Beatles write any music in the Disco genre?
Answer: false

Evidence: Ganymede is a moon of Jupiter. Jupiter is the largest planet in our solar system. The solar system is part of the Milky
↪  Way galaxy.
Question: Is Ganymede in the Milky Way galaxy?
Answer: true

## A.6.4 FEVER

```
Evidence: Segarra served as Military Aide to the Military Governor of Puerto Rico Theodore Roosevelt , Jr. and during World War II
↪    commanded the 65th Infantry Regiment .
Question: Raees (film) features a Pakistani actress in a lead role.
Answer: error

Evidence: With an estimated 92.7 million inhabitants , it is the world 's 14th-most-populous country , and the ninth-most-populous
↪    Asian country .
Question: Vietnam is not the ninth most populous Asian country.
Answer: false

Evidence: Since Trump 's inauguration , Conway has been embroiled in a series of controversies , including using the phrase ``
↪    alternative facts '' , making reference to a `` Bowling Green massacre '' that never occurred , claiming Michael Flynn had the
↪    full confidence of the president hours before he was dismissed , and publicly endorsing commercial products associated with the
↪    president 's daughter Ivanka Trump .
Question: Kellyanne Conway has been embroiled in a series of controversies.
Answer: true

Evidence: The village has around 2000 families . This village is famous for the celebrations of Batukamma festival celebrated during
↪    Dushera.People from near by villages come here to play Batukamma .
Question: Tatum O'Neal had three children with juvenile arthritis.
Answer: error

Evidence: It is being directed by Brian Fee , a storyboard artist on Cars -LRB- 2006 -RRB- and Cars 2 -LRB- 2011 -RRB- .
Question: Cars 3 isn't the third Cars movie Brian Fee has worked on.
Answer: false

Evidence: `` Shut Up '' is a song by English Grime artist and MC Stormzy .
Question: Shut Up is a song.
Answer: true

Evidence: Patrice Loko , French former footballer
Question: The dramatic film The Good German was directed by Steven Soderburgh.
Answer: error

Evidence: Poseidon grossed $ 181,674,817 at the worldwide box office on a budget of $ 160 million .
Question: Poseidon lost $181,674,817.
Answer: false

Evidence: Qui-Gon Jinn is a fictional character in the Star Wars franchise , portrayed by Liam Neeson as the main protagonist of the
↪    1999 film Star Wars : Episode I -- The Phantom Menace '' .
Question: Qui-Gon Jinn is a character in the Star Wars franchise.
Answer: true

Evidence: It is only known from a partial skull and several vertebrae , but comparisons with other species of monitor lizard put its
↪    size between 60 and in length .
Question: The Mormon population has grown significantly since 1980.
Answer: error

Evidence: Python features a dynamic type system and automatic memory management and supports multiple programming paradigms ,
↪    including object-oriented , imperative , functional programming , and procedural styles .
Question: Python lacks a dynamic type system.
Answer: false

Evidence: The series was nominated for four Annie Awards in 2017 , winning three in the categories of Outstanding Achievement in
↪    Character Animation , Character Design , and Storyboarding in an Animated Television/Broadcast Production . Trollhunters is an
↪    American computer-animated fantasy television series created for Netflix by Guillermo del Toro and produced by DreamWorks
↪    Animation and Double Dare You .
Question: Trollhunters is animated.
Answer: true

Evidence: Guillermo Kuschel -LRB- born 1918 -RRB- , Chile-born entomologist living in New Zealand Maximilian Kuschel -LRB- 1851 --
↪    1909 -RRB- , German ornithologist and oologist
Question: Shawn Carlson is American and German.
Answer: error

Evidence: Kutcher subsequently appeared in more romantic comedies , including Guess Who -LRB- 2005 -RRB- , A Lot Like Love -LRB-
↪    2005 -RRB- , What Happens in Vegas -LRB- 2008 -RRB- , and No Strings Attached -LRB- 2011 -RRB- . No Strings Attached is a 2011
↪    American romantic comedy film directed by Ivan Reitman and written by Elizabeth Meriwether .
Question: Ashton Kutcher was not directed by Ivan Reitman.
Answer: false

Evidence: It was formerly called Irian Jaya -LRB- before that West Irian or Irian Barat -RRB- and comprised all of Indonesian New
↪    Guinea .
Question: Papua was formerly called West Iran.
Answer: true
```

## A.7 PROMPTS FOR CALCULATING SCORES

Here, we provide the prompts used to obtain the different scores considered in the different factorizations. We provide the prompts derived for NQ.

## A.7.1  CALCULATING $p(q \mid a_i, p_i)$

Evidence: FIFA World Rankings. Top 20 rankings as of 16 October 2017 Rank Change Team Points Germany 1631 Brazil 1619 Portugal 1446
↪  Argentina 1445 5 Belgium 1333 6 Poland 1323 7 France 1226 8 Spain 1218 9 Chile 1173 10 Peru 1160 11 Switzerland 1134 12 England
↪  1116 13 Colombia 1095 14 Wales 1072 15 Italy 1066 16 Mexico 1060 17 Uruguay 1034 18 Croatia 1013 19 7 Denmark 1001 20 9
↪  Netherlands 931 * Change from 14 September 2017 Complete rankings at FIFA.com
Answer: Germany
Question: who has been ranked no. 1 in the latest football rankings announced by fifa

Evidence: Your Love (The Outfield song). "Your Love" is a song by the English rock band the Outfield, taken from their debut album
↪  Play Deep (1985). The song was penned by the band's guitarist John Spinks.
Answer: English rock band the Outfield
Question: who sings i just want to use your love tonight

Evidence: The Glass Castle (film). Principal photography began on May 20, 2016, in Welch, West Virginia.
Answer: in Welch, West Virginia
Question: where was the movie the glass castle filmed

Evidence: List of nominated members of Rajya Sabha. No. Name Field Affiliation Date of Appointment Date of Retirement Roopa Ganguly
↪  Art Bharatiya Janata Party 04-Oct-2016 03-Oct-2022 Sambhaji Raje Social work Bharatiya Janata Party 07-Jun-2016 03-May-2022
↪  Suresh Gopi Art Bharatiya Janata Party 25-Apr-2016 24-Apr-2022 Subramanian Swamy Economics Bharatiya Janata Party 25-Apr-2016
↪  24-Apr-2022 5 Narendra Jadhav Economics Nominated 25-Apr-2016 24-Apr-2022 6 Mary Kom Sport Nominated 25-Apr-2016 24-Apr-2022 7
↪  Swapan Dasgupta Journalism Nominated 25-Apr-2016 24-Apr-2022 8 K.T.S. Tulsi Law Nominated 25-Feb-2014 24-Feb-2020 9 K. Parasaran
↪  Law Nominated 09-Jun-2012 28-Jun-2018 10 Rekha Art Nominated 27-Apr-2012 26-Apr-2018 11 Sachin Tendulkar Social service
↪  Nominated 27-Apr-2012 26-Apr-2018 12 Anu Aga Business Nominated 27-Apr-2012 26-Apr-2018
Answer: Mary Kom
Question: who was the first lady nominated member of the rajya sabha

Evidence: McChicken. The McChicken is a chicken sandwich sold by the international fast-food chain McDonald's. The sandwich consists
↪  of a toasted wheat bun, a breaded chicken patty, shredded lettuce, and mayonnaise.
Answer: a breaded chicken patty
Question: what is on a mcchicken sandwich from mcdonalds

Evidence: Life of Pi. Life of Pi is a Canadian fantasy adventure novel by Yann Martel published in 2001. The protagonist is Piscine
↪  Molitor "Pi" Patel, an Indian boy from Pondicherry who explores issues of spirituality and practicality from an early age. He
↪  survives 227 days after a shipwreck while stranded on a lifeboat in the Pacific Ocean with a Bengal tiger named Richard Parker.
Answer: Richard Parker
Question: what is the tigers name in life of pi

Evidence: Malware. Malware, short for malicious software, is an umbrella term used to refer to a variety of forms of harmful or
↪  intrusive software, including computer viruses, worms, Trojan horses, ransomware, spyware, adware, scareware, and other
↪  malicious programs. It can take the form of executable code, scripts, active content, and other software. Malware is defined by
↪  its malicious intent, acting against the requirements of the computer user -- and so does not include software that causes
↪  unintentional harm due to some deficiency.
Answer: Malware
Question: the general term for software that is designed to damage disable or steal data is

Evidence: Mum (TV series). Mum Genre Sitcom Created by Stefan Golaszewski Written by Stefan Golaszewski Directed by Richard Laxton
↪  Stefan Golaszewski Starring Lesley Manville Peter Mullan Sam Swainsbury Lisa McGrillis Opening theme Cups by Lulu and the
↪  Lampshades Country of origin United Kingdom Original language (s) English No. of series No. of episodes 12 (to 27 March 2018)
↪  Production Running time 30 minutes Production company (s) Big Talk Productions Distributor ITV Studios Release Original network
↪  BBC Two (2016-present) BBC Two HD (2016-present) Picture format 16: 9 1080i Audio format Stereo Original release 13 May 2016
↪  (2016-05-13) -- present
Answer: Lulu and the Lampshades
Question: who sings the theme tune to mum on bbc2

Evidence: Chess World Cup 2017. The Chess World Cup 2017 was a 128-player single-elimination chess tournament, held in Tbilisi,
↪  Georgia, from 2 to 27 September 2017. It was won by Armenian grandmaster Levon Aronian. This was the second time he had won the
↪  Chess World Cup, 12 years after his first win in 2005.
Answer: Tbilisi, Georgia
Question: where was the world chess tournament 2017 held

Evidence: F Is for Family. T.J. Miller as Randy Kevin Michael Richardson as Rosie, others David Koechner as Robert "Bob Pogo"
↪  Pogrohvich, Frank's obese, chainsmoking boss. Kevin Farley as Babe, Carl, others Gary Cole as Rodger Dunbarton, the owner and
↪  founder of the airlines where Frank and his co-workers work. Joe Buck as Lou Gagliardi, others John DiMaggio as Scoop Dunbarton,
↪  Roger Dunbarton's racist and moronic nephew. Allison Janney as Henrietta Van Horne T.J. Miller as Randy Michael K. Williams as
↪  Smoky
Answer: T.J. Miller
Question: who voices randy in f is for family

Evidence: Las Vegas Stadium. Las Vegas Stadium is the working name for a domed stadium under construction in Paradise, Nevada for
↪  the Las Vegas Raiders of the National Football League (NFL) and the UNLV Rebels football team from the University of Nevada, Las
↪  Vegas (UNLV). It is located on about 62 acres west of Mandalay Bay at Russell Road and Hacienda Avenue and between Polaris
↪  Avenue and Dean Martin Drive, just west of Interstate 15. Construction of the $1.9 billion stadium began in September 2017 and
↪  is expected to be completed in time for the 2020 NFL season.
Answer: Paradise, Nevada
Question: where are they building the new raiders stadium

Evidence: Starbucks. At the time of its initial public offering (IPO) on the stock market in June 1992, Starbucks had 140 outlets,
↪  with a revenue of US $ 73.5 million, up from US $1.3 million in 1987. The company's market value was US $271 million by this
↪  time. The 12% portion of the company that was sold raised around US $25 million for the company, which facilitated a doubling of
↪  the number of stores over the next two years. By September 1992, Starbucks ' share price had risen by 70% to over 100 times the
↪  earnings per share of the previous year.
Answer: June 1992
Question: when did starbucks become a publicly traded company

Evidence: Wipro. At the end of December 31, 2015, its employee strength was 170,664. Abid Ali Neemuchwala was appointed as Wipro's
↪ CEO after T.K. stepped down in early 2016.
Answer: Abid Ali Neemuchwala
Question: who become the ceo of it wipro company in 2016

Evidence: The Next Iron Chef. WINNER: Geoffrey Zakarian Episode 5 6 7 8 Comments Zakarian WIN IN IN CO IN CO WIN WIN The Next Iron
↪ Chef Falkner IN IN WIN IN CO WIN CO OUT Elim: Pressure Chiarello IN CO IN WIN IN IN OUT Elim: Passion Guarnaschelli IN WIN IN IN
↪ IN IN OUT Burrell IN IN IN IN WIN OUT Elim: Risk Samuelsson CO IN IN IN OUT Elim: Storytelling MacMillan WIN IN CO OUT Elim:
↪ Improvisation Hughes IN IN OUT Elim: Ingenuity Irvine IN OUT Elim: Transformation Mendelsohn OUT Elim: Resourcefulness
Answer: Zakarian
Question: who wins the next iron chef super chefs

Evidence: Plant stem. Support for and the elevation of leaves, flowers and fruits. The stems keep the leaves in the light and
↪ provide a place for the plant to keep its flowers and fruits. Transport of fluids between the roots and the shoots in the xylem
↪ and phloem Storage of nutrients Production of new living tissue. The normal lifespan of plant cells is one to three years. Stems
↪ have cells called meristems that annually generate new living tissue.
Answer: Production of new living tissue
Question: what are the main functions of the stem

## A.7.2  CALCULATING $p(q \mid p_i)$

Same as above, but omitting the *Answer* field of the prompt.

## A.7.3  CALCULATING $p(p_i \mid q)$

Question: who has been ranked no. 1 in the latest football rankings announced by fifa
Evidence: FIFA World Rankings. Top 20 rankings as of 16 October 2017 Rank Change Team Points Germany 1631 Brazil 1619 Portugal 1446
↪ Argentina 1445 5 Belgium 1333 6 Poland 1323 7 France 1226 8 Spain 1218 9 Chile 1173 10 Peru 1160 11 Switzerland 1134 12 England
↪ 1116 13 Colombia 1095 14 Wales 1072 15 Italy 1066 16 Mexico 1060 17 Uruguay 1034 18 Croatia 1013 19 7 Denmark 1001 20 9
↪ Netherlands 931 * Change from 14 September 2017 Complete rankings at FIFA.com

Question: who sings i just want to use your love tonight
Evidence: Your Love (The Outfield song). "Your Love" is a song by the English rock band the Outfield, taken from their debut album
↪ Play Deep (1985). The song was penned by the band's guitarist John Spinks.

Question: where was the movie the glass castle filmed
Evidence: The Glass Castle (film). Principal photography began on May 20, 2016, in Welch, West Virginia.

Question: who was the first lady nominated member of the rajya sabha
Evidence: List of nominated members of Rajya Sabha. No. Name Field Affiliation Date of Appointment Date of Retirement Roopa Ganguly
↪ Art Bharatiya Janata Party 04-Oct-2016 03-Oct-2022 Sambhaji Raje Social work Bharatiya Janata Party 07-Jun-2016 03-May-2022
↪ Suresh Gopi Art Bharatiya Janata Party 25-Apr-2016 24-Apr-2022 Subramanian Swamy Economics Bharatiya Janata Party 25-Apr-2016
↪ 24-Apr-2022 5 Narendra Jadhav Economics Nominated 25-Apr-2016 24-Apr-2022 6 Mary Kom Sport Nominated 25-Apr-2016 24-Apr-2022 7
↪ Swapan Dasgupta Journalism Nominated 25-Apr-2016 24-Apr-2022 8 K.T.S. Tulsi Law Nominated 25-Feb-2014 24-Feb-2020 9 K. Parasaran
↪ Law Nominated 09-Jun-2012 28-Jun-2018 10 Rekha Art Nominated 27-Apr-2012 26-Apr-2018 11 Sachin Tendulkar Social service
↪ Nominated 27-Apr-2012 26-Apr-2018 12 Anu Aga Business Nominated 27-Apr-2012 26-Apr-2018

Question: what is on a mcchicken sandwich from mcdonalds
Evidence: McChicken. The McChicken is a chicken sandwich sold by the international fast-food chain McDonald's. The sandwich consists
↪ of a toasted wheat bun, a breaded chicken patty, shredded lettuce, and mayonnaise.

Question: what is the tigers name in life of pi
Evidence: Life of Pi. Life of Pi is a Canadian fantasy adventure novel by Yann Martel published in 2001. The protagonist is Piscine
↪ Molitor "Pi" Patel, an Indian boy from Pondicherry who explores issues of spirituality and practicality from an early age. He
↪ survives 227 days after a shipwreck while stranded on a lifeboat in the Pacific Ocean with a Bengal tiger named Richard Parker.

Question: the general term for software that is designed to damage disable or steal data is
Evidence: Malware. Malware, short for malicious software, is an umbrella term used to refer to a variety of forms of harmful or
↪ intrusive software, including computer viruses, worms, Trojan horses, ransomware, spyware, adware, scareware, and other
↪ malicious programs. It can take the form of executable code, scripts, active content, and other software. Malware is defined by
↪ its malicious intent, acting against the requirements of the computer user -- and so does not include software that causes
↪ unintentional harm due to some deficiency.

Question: who sings the theme tune to mum on bbc2
Evidence: Mum (TV series). Mum Genre Sitcom Created by Stefan Golaszewski Written by Stefan Golaszewski Directed by Richard Laxton
↪ Stefan Golaszewski Starring Lesley Manville Peter Mullan Sam Swainsbury Lisa McGrillis Opening theme Cups by Lulu and the
↪ Lampshades Country of origin United Kingdom Original language (s) English No. of series No. of episodes 12 (to 27 March 2018)
↪ Production Running time 30 minutes Production company (s) Big Talk Productions Distributor ITV Studios Release Original network
↪ BBC Two (2016-present) BBC Two HD (2016-present) Picture format 16: 9 1080i Audio format Stereo Original release 13 May 2016
↪ (2016-05-13) -- present

Question: where was the world chess tournament 2017 held
Evidence: Chess World Cup 2017. The Chess World Cup 2017 was a 128-player single-elimination chess tournament, held in Tbilisi,
↪ Georgia, from 2 to 27 September 2017. It was won by Armenian grandmaster Levon Aronian. This was the second time he had won the
↪ Chess World Cup, 12 years after his first win in 2005.

Question: who voices randy in f is for family
Evidence: F Is for Family. T.J. Miller as Randy Kevin Michael Richardson as Rosie, others David Koechner as Robert "Bob Pogo"
↪ Pogrohvich, Frank's obese, chainsmoking boss. Kevin Farley as Babe, Carl, others Gary Cole as Rodger Dunbarton, the owner and
↪ founder of the airlines where Frank and his co-workers work. Joe Buck as Lou Gagliardi, others John DiMaggio as Scoop Dunbarton,
↪ Roger Dunbarton's racist and moronic nephew. Allison Janney as Henrietta Van Horne T.J. Miller as Randy Michael K. Williams as
↪ Smoky

Question: where are they building the new raiders stadium
Evidence: Las Vegas Stadium. Las Vegas Stadium is the working name for a domed stadium under construction in Paradise, Nevada for
↪ the Las Vegas Raiders of the National Football League (NFL) and the UNLV Rebels football team from the University of Nevada, Las
↪ Vegas (UNLV). It is located on about 62 acres west of Mandalay Bay at Russell Road and Hacienda Avenue and between Polaris
↪ Avenue and Dean Martin Drive, just west of Interstate 15. Construction of the $1.9 billion stadium began in September 2017 and
↪ is expected to be completed in time for the 2020 NFL season.

```
Question: when did starbucks become a publicly traded company
Evidence: Starbucks. At the time of its initial public offering (IPO) on the stock market in June 1992, Starbucks had 140 outlets,
↪  with a revenue of US $ 73.5 million, up from US $1.3 million in 1987. The company's market value was US $271 million by this
↪  time. The 12% portion of the company that was sold raised around US $25 million for the company, which facilitated a doubling of
↪  the number of stores over the next two years. By September 1992, Starbucks ' share price had risen by 70% to over 100 times the
↪  earnings per share of the previous year.

Question: who become the ceo of it wipro company in 2016
Evidence: Wipro. At the end of December 31, 2015, its employee strength was 170,664. Abid Ali Neemuchwala was appointed as Wipro's
↪  CEO after T.K. stepped down in early 2016.

Question: who wins the next iron chef super chefs
Evidence: The Next Iron Chef. WINNER: Geoffrey Zakarian Episode 5 6 7 8 Comments Zakarian WIN IN IN CO IN CO WIN WIN The Next Iron
↪  Chef Falkner IN IN WIN IN CO WIN CO OUT Elim: Pressure Chiarello IN CO IN WIN IN IN OUT Elim: Passion Guarnaschelli IN WIN IN IN
↪  IN IN OUT Burrell IN IN IN IN WIN OUT Elim: Risk Samuelsson CO IN IN IN OUT Elim: Storytelling MacMillan WIN IN CO OUT Elim:
↪  Improvisation Hughes IN IN OUT Elim: Ingenuity Irvine IN OUT Elim: Transformation Mendelsohn OUT Elim: Resourcefulness

Question: what are the main functions of the stem
Evidence: Plant stem. Support for and the elevation of leaves, flowers and fruits. The stems keep the leaves in the light and
↪  provide a place for the plant to keep its flowers and fruits. Transport of fluids between the roots and the shoots in the xylem
↪  and phloem Storage of nutrients Production of new living tissue. The normal lifespan of plant cells is one to three years. Stems
↪  have cells called meristems that annually generate new living tissue.
```

## A.8   ERROR ANALYSIS

In addition to discussion of SITUATEDQA in Section 5.5, which can be considered a tail of NQ dataset, we also perform error analysis of HOTPOTQA and STRATEGYQA; see examples in Table 4.

Questions in HOTPOTQA fall into two categories: comparison (e.g., "Which was fought earlier in our nation's history, the Seven Days Battles or the Battle of Manila?") and bridge entity questions (e.g., "Which American professional poker player also starred in the 2015 movie "Extraction"?"). While performance of $OB_{Gold}$ is comparable for both types, the bridge questions appear particularly difficult for both CB and $OB_{Google}^{a|q,p}$ configurations: for $OB_{Google}^{a|q,p}$ we observe EM of 21% for bridge vs 47% for comparison questions. We hypothesize that this is because comparison questions always have access to the correct answers in the prompt through the question itself, even if retrieval fails. Having a correct answer in the prompt makes the task easier for the model. In contrast, for bridge questions the answer needs to be generated "from scratch" if it is not present in retrieved evidence.

Manual error analysis of questions where Google retrieval, $OB_{Google}^{a|q,p}$, and CB fail simultaneously suggests that there are several sources of error. One is Google Search's recency bias, some questions require older documents; next are failures of Exact Match as an accuracy metric (e.g., 'rookie of year' vs 'national basketball associations rookie of year', 'nandrolone' vs 'anabolic–androgenic steroid aas'); finally, we also observe data annotations errors (multiple gold answers are possible instead of a single one provided or the gold answers are wrong). Perhaps surprisingly, on a small subset of questions that we manually checked, we observe the latter two to be the prevalent causes of errors.

Therefore, the measured accuracy of 21% on bridge questions is likely a lower-bound of $OB_{Google}^{a|q,p}$ performance. In fact, we believe that the comparison questions present a more difficult challenge to the language models, as they often require numeric reasoning.

The corner cases of STRATEGYQA include questions where Google Search either entirely fails to retrieve evidence (e.g., "Could a sloth hypothetically watch an entire episode of Scrubs underwater?") or it retrieves relevant evidence that contradicts reasoning provided by annotators (e.g., "Can vitamin C rich fruits be bad for health?") (note also that some of the questions are subjective, with different points of view being equally valid). Manual analysis of the results also suggests that most of the questions in the dataset require iterative retrieval and reasoning, where the sub-questions for the individual steps cannot be obtained from the original question through simple modifications, but require reformulation. Hence, whereas retrieved evidence is generally on topic, it is not accurate enough to support fine-grained reasoning required to answer the questions.

**Example 1** HOTPOTQA : $OB_{Google}^{PoE}$ **is correct**

**Q:** Which American professional poker player also starred in the 2015 movie "Extraction"?

**A** $OB_{Google}^{PoE}$: `dan bilzerian`

**A CB:** `chris ferguson`

**A Gold:** `dan bilzerian`

**Type**: bridge

**Evidence@1** `Dan Bilzerian` - Bio, Net Worth, Height Famous Births
Deaths. Dan Bilzerian Bio, Net Worth, Height ORIGIN Dan Bilzerian is an
American professional poker player, . . .

**Evidence@2** `Dan Bilzerian` Net Worth (As of Oct. 2021) - Celebinsidr.com.
He has earned most of his money by playing poker. . . .

**Evidence@3** `Dan Bilzerian` - Biography of a Poker Playing Playboy - Online `Dan Bilzerian` :
Poker-Playing Playboy `Dan Bilzerian` is your typical playboy. He surrounds himself with
beautiful women, fast cars, and just about anything his heart desires..

**Example 2** HOTPOTQA : $OB_{Google}^{PoE}$ **fails**

**Q:** Who is the American singer-songwriter, who won an award for
Best Female Video at the 2009 MTV Video Music Awards,
and wrote a song for the ATT Team USA Soundtrack?

**A** $OB_{Google}^{PoE}$: `colbie caillat`

**A CB:** `taylor swift`

**A Gold:** `taylor swift`

**Type:** bridge

**Evidence@1** `Colbie Caillat` - Wikipedia. In August 2009 she released Breakthrough, ..
She was also part of the group
that won Album of the Year at the 2010 Grammy Awards
for her featured vocals and writing on `Taylor Swift` 's Fearless album.. . .

**Evidence@2** `Colbie Caillat` - Wikipedia. She also recorded a French translated version of this song. . . .

**Evidence@3** The Songs - `Taylor Swift` Switzerland. Then there is the absolute sadness.
The sadness of losing this person, losing all the memories, and the hopes you had for the future...

**Example 3** HOTPOTQA : $OB_{Google}^{PoE}$ **fails**

**Q:** Which was fought earlier in our nation's history, the Seven Days Battles or the Battle of Manila?

**A** $OB_{Google}^{PoE}$: `battle of manila`

**A CB:** `seven days battles`

**A Gold:** `seven days battles`

**Type:** comparison

**Evidence@1** `The Seven Days Battles` American Battlefield Trust. .. set back almost a year.
By late afternoon on May 31, 1862. . .

**Evidence@2** `Battle of Manila` (1898) - Wikipedia. Army private fired the first shot
at a Filipino revolutionary soldier and Filipino revolutionary forces returned fire. . . .

**Evidence@3** `Battle of Manila` (1945) - Wikipedia. The city's capture was marked as General Douglas
MacArthur's key to victory in the campaign of reconquest. . . .

**Example 4** HOTPOTQA : $OB_{Google}^{PoE}$, CB both **fail**

**Q:** What North Carolina native did Danja produce songs for?

**A** $OB_{Google}^{PoE}$: `justin timberlake`

**A CB:** britney spears

**A Gold:** j cole

**Type:** bridge

**Evidence@1** Producer Danja Explains Why `Justin Timberlake` s "Filthy" Doesní ....
I caní say that "Filthy" was described in the trailer,..

**Evidence@2** Producer Danja Explains Why `Justin Timberlake` s "Filthy" Doesní ....
This is what we do. What were some of those parameters that he gave you?

**Evidence@3** Danja production discography - Wikipedia. "Yeah Yeah You Would" (feat. Grace Jones) -
06. "Hate You Now" - 14. "Hello Good Morning" (feat. T.I.)
Keri Hilson - No Boys Allowed[edit] * 07. "Toy Soldier" (co-written with Keri) * 16.

**Example 5** STRATEGYQA : $OB_{Google}^{PoE}$ **fails**

**Q:** Does Santa Claus work during summer?

**A** $OB_{Google}^{PoE}$: True

**A CB:** False

**A Gold:** False

**Evidence@1** What Does Santa Claus Do In The Summer? — Mystic Christmas Blog. Mrs. ..
During the Summer Santa and his Elves also `continue their worldwide charity work`
along with fighting the forces of evil around the world. ..

**Evidence@2** What Does Santa Claus Do In The Summer? — Mystic Christmas Blog.
What Does Santa Claus Do In The Summer? ..
However the term Christmas in July was started by Santa Claus.
He `begins his preparations` for the monumental Christmas season ..
What it's really like to be a professional Santa Claus - Business Insider. ..

Table 4: Examples of questions, CB and OB answers together with retrieved evidence from HOTPOTQA
and STRATEGYQA datasets.

