# OpenReview forum: "Internet-augmented language models through few-shot prompting for open-domain question answering"
_ICLR.cc/2023/Conference — Submitted to ICLR 2023_

### Official Review · Reviewer_rQxP · 2022-10-24

**Confidence:** 4
**Correctness:** 3
**Technical Novelty And Significance:** 2
**Empirical Novelty And Significance:** 2
**Recommendation:** 6

**Clarity, Quality, Novelty And Reproducibility:**

The most parts of the paper are clear, and easy-to follow given its well-written style. But, an exception is that the method of product-of-experts (PoE) is clearly described. Also, the equation for direction inference needs to be checked: should the max be used instead of the summation?

The proposed method of the few-shot “in-context” learning for internet-augmented model in the GPT3-style autoregressive language models is novel, which has not been explored yet.


**Strength And Weaknesses:**

Strengths

- This paper is well written and easy to follow the major idea and content.
- The experiment results are solid, showing consistent improvements comparing to the huge closed book model.
- The proposed methods are simple but valuable to be explored, particularly presenting a new approach for the few-shot “in-context” learning in retrieval augmented language model under the autoregressive transformer.

Weaknesses

- Only google search results are used for retrieval. It is not clear whether the improvement come from the high quality of google’s retrieved contents or from the retrieval-augmented method. Other collection sources for retrieval such as Wikipedia or the training corpus of Gopher 280B also need to be used for evaluation, in addition to Google’s results on internet. In other words, the authors further need to whether the improvements over CB (in Table mainly come from by keeping or perfectly memorizing the corpus used for pretraining CB, or whether it comes from the use of new but well-matched corpus that is NOT seen when pretraining CB.
- The proposed method is applied to the autoregressive transformer using the “in-context” learning. But, it is not clear how the method is extended to the encoder-decoder model like T5.
- Evaluation datasets are limited. KILT or other knowledge-intensive datasets need to be further used for evaluation.
- What is the advantage comparing to other few-shot learning of retrieval-augmented models, such as:
Gautier Izacard, Patrick Lewis, Maria Lomeli, Lucas Hosseini, Fabio Petroni, Timo Schick, Jane Dwivedi-Yu, Armand Joulin, Sebastian Riedel, Edouard Grave, Few-shot Learning with Retrieval Augmented Language Modelsm 2022:

   How the proposed method is complementary applied to the above few-shot learning method?



**Summary Of The Paper:**

This paper proposes the method of few-shot prompting for internet-augmented language models consisting of three steps: 1) retrieval, which searches top 50 paragraphs using Google search, 2) prompting, where four prompts are applied on each of top 50 retrieved paragraphs to generate 200 candidate answers in total, 3) reranking, which reranks these candiate answers. In particular, authors present various methods for the “reranking” step – RAG-style inference, noisy channel inference, and PoE. Experiment results on NQ, HotpotQA, FEVER, and StrategyQA show that the proposed few-shot internet-augmented LMs make further improvements, comparing to the closed book model, Gopher 280B, under both no-reranking and reranking settings. When evaluating the performance with respect to inference time, the proposed few-shot prompting produces better performances than the closed book model, when using similar inference time (i.e. FLOPS).

The key contribution of the paper is the simple but novel few-shot prompting for using internet search to improve the large language model under the autoregressive Transformer architecture, thereby confirming that that retrieval-augmented models are more promising and effective than the huge closed book model (i.e., Gopher 280B).


**Summary Of The Review:**

Overall, this paper is well-written and easy to follow. The proposed idea of few-shot prompting for internet-augmented language models is novel, interesting, and timely proper in the literature of the autoregressive language model. Experiment results are solid, showing improvements over the baseline huge Gopher 280B, and Gopher’s performance can be achieved by much smaller model. The concluded direction of reconsidering towards huge pretrained models reasonably run from the experiments in the paper.

Further comments:

- In the right of Figure 3, CB 280B is only plotted. But, it would be better or required to plot the results of other smaller CB models.
- In Table 1, the results may rely on the performance of the retrieval. In addition to using other resources like Wikipedia, can we control the retrieval performance by randomly putting noise irrelevant passages and check the degree of the performance from which the final QA performance start to decrease?

---

> ### Author Response · Authors · 2022-11-18
> **Author response**
>
> We thank the reviewer for their comments.
>
> > Only google search results are used for retrieval.
>
> In Appendix A.5 we have shown that our method works even when conditioning on Wikipedia evidence.
>
> > Whether it comes from the use of new but well-matched corpus that is NOT seen when pretraining CB.
>
> To test for this, we have included in Section 5.5 an experiment that uses questions about questions that can only be answered with knowledge of 2021 events, which is the training cut-off for our CB models (hence our training data do not contain enough information to answer these questions). Indeed, in Table 2 we show that we can use our method to account for this lack of information in the weights of the model.
>
> > Evaluation datasets are limited.
>
> We have evaluated our method on 5 datasets, FEVER, StrategyQA, HotpotQA, NQ and Situated QA (3 of these are in fact already part of KILT). Moreover, Appendix A3 provides results on a subset of BEIR benchmark.
>
> > The proposed method is applied to the autoregressive transformer using the “in-context” learning. But, it is not clear how the method is extended to the encoder-decoder model like T5.
>
> This is a great question. There is nothing in our method that makes us believe that this wouldn’t work with T5-like models to the degree that it has been shown that encoder-decoder models also have good in-context learning abilities. However, we agree that in the future we need to verify that.
>
>
> > What is the advantage comparing to other few-shot learning of retrieval-augmented models
>
> ATLAS uses few-shot learning, i.e., uses a limited number of data to fine-tune the model. On the other hand we study in-context learning and hence do not require any training, which can be tricky if working with large models powered by APIs. As such, a potential advantage is that when training is not feasible, one can use our approach to condition on external evidence. Beyond that, these models train cross-attention to consider evidence, whereas we rely on the strong pre-trained capabilities of self-attention. An interesting experiment would be combining the two, wherein evidence is both provided to the self-attention and to the cross-attention.
>
> > In the right of Figure 3, CB 280B is only plotted. But, it would be better or required to plot the results of other smaller CB models.
>
> FLOPs for 1B CB  is ~10^12 whereas QA performance is ~ 5 exact match (from Figure 2 right). FLOPs for 7B CB  is ~10^13 whereas QA performance is ~12 exact match (again from Figure 2 right).
>
> Overall, the QA performance of smaller CB models is really low, while their FLOPs are roughly the same as the equivalent in size open-book models with only 1 evidence, which however perform significantly better.

---

> > ### Author Response · Authors · 2022-11-18
> > **Robustness of system as a function of retrieval performance**
> >
> > Thank you for suggesting this interesting experiment.
> >
> > This is what we did: We report performance of $OB^{a|qp}_{Google}$  when considering 20 passages starting from the top20 passages and each time swapping out a passages replacing them with passage from neighbours ranked lower (between top30 and top50).
> > Results are bellow as percentage of noise included: 0% means initial performance of top20 passages, 100% means performance when considering the 20 neighbours ranked 30th to 50th, anything else means swapping % of top neighbours with lower ranked neighbours.
> >
> > Overall, we do observe a small drop -- ignoring the top30 neighbours results in a 3.7% drop of performance.
> >
> > |     Noise level  | Exact match|
> > |----------|-------|
> > |0% | 33.7 |
> > |10%  | 34.1|
> > |20 % | 34.0|
> > |30% | 33.2 |
> > |40 % | 32.6|
> > |50 % | 32.1 |
> > |60% | 31.6|
> > |70% | 31.3|
> > |90% | 31.2|
> > |100 % |30.0|

---

### Official Review · Reviewer_YZVR · 2022-10-24

**Confidence:** 3
**Correctness:** 3
**Technical Novelty And Significance:** 2
**Empirical Novelty And Significance:** 3
**Recommendation:** 6

**Clarity, Quality, Novelty And Reproducibility:**

Overall, the novelty and quality of this paper are good. But the clarity about different reranking methods is a concern. This paper needs to elaborate more on each reranking method and the difference between them. Currently there's just one-sentence description for each method. Especially for Product-of-Experts (PoE), which is the most effective one. How to combine those probabilities is not clear.
Moreover, the reproducibility is limited since it's only tested on the Gopher model, which seems not publicly available either through API or model weights.

**Strength And Weaknesses:**

Strength:
1. This paper utilizes web-scale information through google search for downstream tasks like open-domain QA.
2. It proposes 4 methods to rerank the generated answers through LM produced probability without fine-tuning.
3. The proposed methods achieve effective results on 4 downstream datasets. This paper also provides analysis on the model scale and inference time, and the ability to answer questions grounded in different dates.

Weakness:
1. The method is only tested on the Gopher model. It would make this paper much stronger if it's also tested on other large-scale LMs such as OPT and GPT-3.
2. More experiments should be done: How does the number of shots k affect the model performance? Can we improve the results by simply increasing k? Another baseline is to simply combine multiple paragraphs into one paragraph and pass into the model instead of generate different answers using each paragraph separately. In this case we hope the LM can produce more accurate answers by joint inference over multiple paragraphs.

**Summary Of The Paper:**

This paper mainly tackles open-domain question answering. It uses Google search to obtain web information to prompt large-scale language model for answer generation.

**Summary Of The Review:**

Although this paper has weaknesses on insufficient experiments, it has contributions in pushing the research about large-scale LM inference and its combination with web-scale knowledge. Thus my recommendation is marginally above the acceptance threshold.

---

> ### Author Response · Authors · 2022-11-18
> **Author response**
>
> > The method is only tested on the Gopher model. It would make this paper much stronger if it's also tested on other large-scale LMs such as OPT and GPT-3.
>
> We have in fact included results with the OPT model. See Appendix A4.
>
> > Moreover, the reproducibility is limited since it's only tested on the Gopher model, which seems not publicly available either through API or model weights.
>
> Please see our discussion on Appendix A4 where we have used our prompts with the OPT model confirming the same trend between closed- and open-book model.
>
> > More experiments should be done: How does the number of shots k affect the model performance? Can we improve the results by simply increasing k? Another baseline is to simply combine multiple paragraphs into one paragraph and pass into the model instead of generate different answers using each paragraph separately. In this case we hope the LM can produce more accurate answers by joint inference over multiple paragraphs.
>
> These are great suggestions. However, both of these require the model to be able to handle longer context length. Currently, our language model, inline without other LLM out there, can only handle up to 2048, which means we can neither increase the number of shots or keep the same number of shots but with the combined multiple passages. However, we agree with the reviewer that these are great ways to further improve results and we believe that perhaps increasing the context length of models would further improve results of our method.

---

> > ### Comment · Reviewer_YZVR · 2022-12-07
> > **Re: author response**
> >
> > Dear authors,
> >
> > Thanks for your response. However, I still think that the results of OPT are **not** sufficient. Currently, the paper only contains results of OPT-6B on the classification task, and it's even unclear which classification task the numbers are referring to. It's important to have OPT results over all the different model sizes and the results in Table 1. Otherwise, it's unclear how the findings on the Gopher model can be transferred to other large-scale language models.

---

### Official Review · Reviewer_z3mQ · 2022-10-25

**Confidence:** 4
**Correctness:** 4
**Technical Novelty And Significance:** 2
**Empirical Novelty And Significance:** Not applicable
**Recommendation:** 3

**Clarity, Quality, Novelty And Reproducibility:**

The paper has good clarity and easy to understand, but the motivation is unclear and the novelty is limited.

**Strength And Weaknesses:**

Strengths:
* The paper is well-written and easy to understand.
* Plenty of empirical discussion and study.

Weakness:
* The motivation of this work is unclear. It reads like the most significant contribution is using Google search as the information retriever compared with previous work using other search engines on Wikipedia, etc.
* The technical contribution is incremental, without significant novelty. The 3 proposed methods can all be summarized as hyperparameter/model tuning on the retrieval-based LMs, i.e., tuning the search engine, input prompting, and ensemble strategy.
* Although the authors have a lot of discussions on their empirical results, they don't finally deliver impressive insights or conclusions, besides using Google search and prompting performs better.

**Summary Of The Paper:**

This paper proposes to improve the use LLMs on open-domain QA tasks, via searching up-to-date information from internet, few-shot prompting, and inference-time computing with the searched-out information. Specifically,

(1) Retrieve information from Google Search.

(2) Input a few existing examples as the prefix to prompt LLM.

(3) In inference time, re-rank the answers with different retrieved paragraphs as evidence.

**Summary Of The Review:**

Although the paper is well-written and with plenty of experiments and discussions, the proposed approach is not well-motivated and doesn't have significant originality and looks just like training a model towards the benchmark by tuning some modules in a retrieval-based LM and using a very large LM as the backbone.

---

> ### Author Response · Authors · 2022-11-19
> **Author response**
>
> We would like to thank the reviewer for their comments.
>
> > The technical contribution is incremental, without significant novelty. The 3 proposed methods can all be summarized as hyperparameter/model tuning on the retrieval-based LMs, i.e., tuning the search engine, input prompting, and ensemble strategy.
>
> Respectfully, but this is an unfair characterization of the work. In some sense a lot of (good) work out there can be "summarized as hyperparameter/model tuning", and this is particularly true for work using LLMs and prompting.
>
> Our overall position is that given the paradigm shift toward LLM, we do need to invest effort to systematically research how to make them more useable and that is what this paper is about. Bellow is a summary of our findings which we believe have a sense of novelty:
>
> * We show that LLMs can be few-shot prompted to become arbitrary scores (see Section 3.3). There is very little work showing that something like this is indeed possible and works (Izacard and Grave 2021 only show this for ranking passages after finetuning models whereas Sachan et all 2022 (which went on arxiv in June) do a concurrent work to ours making the same point)
> * We show that smaller open-book models can in fact surpass larger closed-book models.
> * We show that we can condition virtually any language model to new information using prompting, something that should minimally act as an easy-to-implement way for anyone working with LLLs without the need for extra fine-tuning.
>
> We hope that the reviewer would accept that some of these points we are making are not all that obvious.

---

### Official Review · Reviewer_H18i · 2022-10-29

**Confidence:** 4
**Correctness:** 3
**Technical Novelty And Significance:** 1
**Empirical Novelty And Significance:** 1
**Recommendation:** 5

**Clarity, Quality, Novelty And Reproducibility:**

Clarity: I think the paper is easy to understand with the exception of an important section (3.3) where the re-ranking is discussed. I think the paper could also benefit from a discussion on how to close the performance gap.

Quality: The paper does a good job in the experimental setup to show us that open-book models is better than closed-book models. Although useful, this is well-known in the community. I still think this is a high quality work with very limited novelty

Novelty: The novelty of the paper is very lacking. Using google search, static prompting are things that has been explored. I also think the paper could have tried other dynamic prompting approaches. The re-ranking section was not motivated well and the re-rankers itself is not novel.

Reproducibility: Unfortunately, very few institutions have the resources to reproduce results on 280B LLMs. However, I appreciate the experiments on the open-source OPT-6B models.

**Strength And Weaknesses:**

**Strengths**

- The proposed approach of the paper is simple. I also agree with the conclusion that instead of racing towards building the largest model, we should carefully explore the few-shot capabilities of mid-sized models.
- Apart from the short motivation of the re-ranking methods, the paper is well-written and readable

**Weaknesses**

- My main concern about the paper is its very limited technical novelty. It uses Google search for retrieval and a set of static prompts (per dataset) to query the LLMs. It proposes three re-ranking methods (without much motivation). Using Google search for retrieval and static prompting has been explored before. There also is a lot of work that explores re-ranking. Although the paper claims that obtaining probabilities/scores from LMs have not been explored, there exist works such as Izacard and Grave 2021, ART (Sachan et al 2022) that use scores from LMs.
- The paper also does not explore the easy-to-try extension of prompting. For example, retrieving similar questions w.r.t a given question. This is easy to try and might work better and I would argue is a more practical prompting strategy than the current strategy of designing dataset-specific prompts.
- Another concern regarding the paper that I have is that most of the results are pretty well-known - e.g. performance of open-book models are better than parametric models, and Google search is better. One interesting result was Gopher 7B + retrieval was better than Gopher 280B without retrieval for NQ and HotpotQA. However, this result does not transfer to reasoning benchmarks, so I am not sure if this result is widely applicable.
- Can you explain the product-of-experts and noisy-channel re-ranking in more detail? It would be also good to know the motivation behind applying this.
- Even though open-book models perform better than closed-book models the overall results when compared to SoTA performance is quite lacking. For example (55.9 v/s 38.4 in NQ, 65.2 v/s 30.3 in HotpotQA). Although it is not required for a paper to achieve state-of-the-art for publication, but I couldn't find any discussion of this wide gap in results and what could be done to fix it. I think the paper would benefit a lot if a section is added which discusses how can this gap be possibly closed.
- Table 1 shows the retrieval performance of Google search and the text in section 5.1 compares that to DPR and MDR. Can you also report the retrieval performance wrt more recent (and better) retrievers such as contriever, etc.
- [Minor] The abbreviation LSLM is first introduced in Section 3 and used in a few places later, but it is unclear what it stands for. Large Semiparametric Language Models?
- [Minor] The second paragraph of Sec A.4 seems to be out of place.

**Summary Of The Paper:**

This paper proposes a simple semiparametric few-shot prompting model for open-domain question answering. For the retriever component of the model, the model uses google search instead of Wikipedia. To account for the long length of documents, each doc is broken into a list of paragraphs where each paragraph is a sequence of 6 sentences. Next, the paragraphs are re-ranked based on Tf-idf similarity within the document.

The prompts for each task are static (i.e. does not change for each query) and consist of 15 (evidence, question, answer) tuples. No parameter training is required.

Another contribution of the paper is to demonstrate that performance of these models can be improved by increasing computing at inference time, i.e., by sampling multiple answers from the model and re-ranking them by the probabilities computed by the model. The paper explores three kinds of ranking - (i) directly estimating p(a|q) by marginalizing an evidence latent variable, (ii) estimating the joint probability of answer and question given the evidence, and (iii) Product-of-Experts which combines (multiplies?) different probabilities from the model. All the probabilities are estimated directly from the model except for p(evidence | question) which is estimated by normalizing the tf-IDF score. The paper should definitely write more to motivate each of the re-ranker, currently, the writing of this part is too short, and especially this is one of the salient contributions of the paper.

The underlying LM used is Gopher LM with varying number of model parameters (44M, 117M, 400M, 1B, 7B, 280B).

The model is tested on 3 open-domain QA datasets (NQ, HotpotQA, StrategyQA)  and a fact-verification dataset (Fever). The proposed semiparametric model outperforms its corresponding closed-book counterparts. However, the best performance of the models is far behind the state-of-the-art (which is surprising) and I thought the paper did a bad job of explaining this difference in performance and what could be done to fix it.

**Summary Of The Review:**

Given the lack of novelty, and the wide gap in performance w.r.t SoTA and also the lack of discussion addressing that result, I am unfortunately tending towards rejecting the paper.

---

> ### Author Response · Authors · 2022-11-18
> **Author response**
>
> > My main concern about the paper is its very limited technical novelty.
>
> While all components have been explored before (something that is probably true for the majority of published work in LLM research today), we are not aware of any work that is putting these components together. Moreover, to the best of our knowledge this is the first time that we see that we can use the LLMs input (i.e., prompt) as a way to keep the model up-to-date with new information. While this is conceptually simple, it opens up the space towards more realistic uses of these models.
>
> > Another concern regarding the paper that I have is that most of the results are pretty well-known - e.g. performance of open-book models are better than parametric models, and Google search is better.
>
> Controlling the model size, open-book models are hands-down better and this is definitely novel. However, our paper shows a number of results that were not known before.
> * First we show that LLMs can be few-shot prompted to become arbitrary scores (see Section 3.3).  Izacard and Grave 2021 only show this for ranking passages after finetuning whereas Sachan et all (who is indeed finding something similar) is really concurrent work to ours as it appeared in Arxiv in June.
> * Second, we show that smaller open-book models can in fact surpass larger closed-book models.
> * Finally, we show that we can condition virtually any language model to new information using prompting, something that should minimally act as an easy-to-implement way for anyone working with LLLs without the need for extra finetuning.
>
> > The paper also does not explore the easy-to-try extension of prompting. For example, retrieving similar questions w.r.t a given question.
>
> We have indeed experimented with these without significant improvements across all our datasets, hence decided to leave out from our discussed experiments.
>
> >  Even though open-book models perform better than closed-book models the overall results when compared to SoTA performance is quite lacking.
>
> Yes, the reviewer is absolutely right and we mention this ourselves in the discussion of results in Section 5.2 as well in the Limitations section. As we mention in the paper, first we find that multi-hop questions challenge our system particularly. We expect that ``learning to search'' approaches could boost performance of the overall system as well as decomposing complex queries into simpler sub-queries, a research avenue that has flourished while this paper was under review!
>
> > The abbreviation LSLM is first introduced in Section 3
>
> We will change to LLM
>
> > The second paragraph of Sec A.4 seems to be out of place.
>
> Thanks for pointing this (and we appreciate the fact that the reviewer also considered the Appendix)
>
> > Table 1 shows the retrieval performance of Google search and the text in section 5.1 compares that to DPR and MDR.
>
> Contriever achieves 82.1 recall@100 (from Table 1 of Contriever paper) whereas we achieve 85 recall@50.
>
> > Can you explain the product-of-experts and noisy-channel re-ranking in more detail? It would be also good to know the motivation behind applying this.
>
> We few-shot prompt the LLM to create different scorers, specifically we create p(a | p), p(q | a, p), p(q | p), p(p | q). For each answer A generated given a question Q and a answer A we calculate the above scores, and together with the  p(a | q,  p) obtained from the QA system, we combine these using a log-linear model. We refer to this as product-of-experts. The motivation for this model is that it’s basically a combination of all the scores that we had calculated for the noisy-channel, the direct inference and the RAG-style model. On the other hand, the noisy-channel essentially factorizes p(a|q, p) such that to include a term about how well a given question can be reconstructed  give a passage (and an answer). The motivation for using the noisy-channel is that we can mitigate problems of “explaining” and by including a term about generation of the question, the passage cannot be ignored anymore as the only way to generate a reasonable question is to pay attention to the passage.

---

> > ### Comment · Reviewer_H18i · 2022-12-05
> > **Re: author response**
> >
> > Dear authors,
> >
> > Thank you for your response.
> >
> > > The paper also does not explore the easy-to-try extension of prompting. For example, retrieving similar questions w.r.t a given question.
> > We have indeed experimented with these without significant improvements across all our datasets, hence decided to leave out from our discussed experiments.
> >
> > I think it will be still interesting to add those results to the paper. For example, this paper (https://arxiv.org/abs/2101.06804) finds that finding more informative prompts for questions is important. It would be interesting to also know more about your findings.
> >
> > > We few-shot prompt the LLM to create different scorers, specifically we create p(a | p), p(q | a, p), p(q | p), p(p | q). For each answer A generated given a question Q and a answer A we calculate the above scores, and together with the p(a | q, p) obtained from the QA system, we combine these using a log-linear model. We refer to this as product-of-experts. The motivation for this model is that it’s basically a combination of all the scores that we had calculated for the noisy-channel, the direct inference and the RAG-style model. On the other hand, the noisy-channel essentially factorizes p(a|q, p) such that to include a term about how well a given question can be reconstructed give a passage (and an answer). The motivation for using the noisy-channel is that we can mitigate problems of “explaining” and by including a term about generation of the question, the passage cannot be ignored anymore as the only way to generate a reasonable question is to pay attention to the passage.
> >
> > Thank you, please add this explanation to the paper.
> >
> > Considering the fact that Sachan et al 2022 is concurrent work (thanks for letting me know) and reading the above responses, I am increasing my original score. Thank you.

---

### Decision · Program_Chairs · 2023-01-20

**Decision:**

Reject

**Justification For Why Not Higher Score:**

Reviewers have expressed concerns on regarding the limited novelty of the method and insufficient experiments (e.g., on more LMs, more prompting methods), as well as the presentation that lack clarity and details.

**Justification For Why Not Lower Score:**

An interesting problem and the proposal of incorporating Google search for retrieval.

**Metareview: Summary, Strengths And Weaknesses:**

The paper proposes a retrieval-augmented prompting approach for open-domain question answering. In particular, the retriever is based on Google search (as opposed to Wikipedia as in previous work). The retrieved documents are reorganized to deal with the excessive length. The paragraphs are then reranked, based on which answers are produced. Different ranking methods are discussed. Experiments are done on the Gopher language models. While the reviewers found the Google-based retrieval interesting, they have also expressed a few concerns regarding the limited novelty of the method and insufficient experiments (e.g., on more LMs, more prompting methods), as well as the presentation that lack clarity and details.